# Nucleotide-level linkage of transcriptional elongation and polyadenylation

Joseph V Geisberg[1†], Zarmik Moqtaderi[1†], Nova Fong[2], Benjamin Erickson[2], David L Bentley[2], Kevin Struhl[1*]

[1]Department of Biological Chemistry and Molecular Pharmacology, Harvard Medical School, Boston, United States; [2]RNA Bioscience Initiative, Department of Biochemistry and Molecular Genetics, University of Colorado School of Medicine, Aurora, United States

**Abstract** Alternative polyadenylation yields many mRNA isoforms whose 3′ termini occur disproportionately in clusters within 3′ untranslated regions. Previously, we showed that profiles of poly(A) site usage are regulated by the rate of transcriptional elongation by RNA polymerase (Pol) II (Geisberg et al., 2020). Pol II derivatives with slow elongation rates confer an upstream-shifted poly(A) profile, whereas fast Pol II strains confer a downstream-shifted poly(A) profile. Within yeast isoform clusters, these shifts occur steadily from one isoform to the next across nucleotide distances. In contrast, the shift between clusters – from the last isoform of one cluster to the first isoform of the next – is much less pronounced, even over large distances. GC content in a region 13–30 nt downstream from isoform clusters correlates with their sensitivity to Pol II elongation rate. In human cells, the upstream shift caused by a slow Pol II mutant also occurs continuously at single nucleotide resolution within clusters but not between them. Pol II occupancy increases just downstream of poly(A) sites, suggesting a linkage between reduced elongation rate and cluster formation. These observations suggest that (1) Pol II elongation speed affects the nucleotide-level dwell time allowing polyadenylation to occur, (2) poly(A) site clusters are linked to the local elongation rate, and hence do not arise simply by intrinsically imprecise cleavage and polyadenylation of the RNA substrate, (3) DNA sequence elements can affect Pol II elongation and poly(A) profiles, and (4) the cleavage/polyadenylation and Pol II elongation complexes are spatially, and perhaps physically, coupled so that polyadenylation occurs rapidly upon emergence of the nascent RNA from the Pol II elongation complex.

**\*For correspondence:**
kevin@hms.harvard.edu

†These authors contributed equally to this work

**Competing interest:** The authors declare that no competing interests exist.

## Editor's evaluation

Geisberg et al. show, in yeast and human cells, a nucleotide-level relationship between the transcriptional elongation rate and the polyadenylation profile. This suggest that the cleavage/polyadenylation and Pol II elongation complexes are spatially, and perhaps physically coupled so that polyadenylation occurs rapidly upon emergence of nascent RNA from the Pol II elongation complex. Furthermore, the GC-content of sequences downstream of poly(A) clusters influences 3′isoform cluster profiles by slowing down elongation, allowing more time for the 3'-cleavage complex to find the poly(A) site. These findings contribute new information on how the transcription machinery determines which poly(A) site are utilized at the end of genes.

## Introduction

The 3′ ends of eukaryotic mRNAs are generated during the process of transcriptional elongation by cleavage of the nascent transcript downstream of the coding region followed by addition of a poly(A) tail (*Proudfoot et al., 2002*; *Tian and Manley, 2013*; *Tian and Manley, 2017*; *Kumar et al., 2019*).

Formation of 3′ ends is mediated by a multiprotein cleavage/polyadenylation (CpA) complex that performs both steps. Alternative polyadenylation gives rise to many 3′ mRNA isoforms differing by the position at which the poly(A) tail is added. The poly(A) profile of a typical yeast gene has ~50 mRNA isoforms with 3′ endpoints occurring within an 'end zone' of ~200 nt (*Ozsolak et al., 2010*; *Moqtaderi et al., 2013*; *Pelechano et al., 2013*). The 3′ untranslated region (3′ UTR) is a modular entity that is sufficient to determine the poly(A) profile (*Lui et al., 2022*). Although mRNA isoforms with neighboring 3′ ends usually have similar properties, they can differ dramatically with respect to mRNA stability, structure throughout the 3′ UTR, and association with Pab1, the poly(A)-binding protein (*Geisberg et al., 2014*; *Moqtaderi et al., 2018*).

Although polyadenylation occurs at numerous sites within the 3′ UTR, it rarely occurs within coding regions (*Moqtaderi et al., 2013*) and introns (*Berg et al., 2012*), even though these are usually much larger. This apparently paradoxical observation has implications for the specificity and mechanism of the CpA machinery, and hence, the poly(A) profile. Polyadenylation in yeast cells is associated with a degenerate sequence motif consisting of a long AU-rich stretch, followed by short U-rich sequences that flank several A residues immediately downstream of the cleavage site (*Guo and Sherman, 1996*; *Moqtaderi et al., 2013*). It has been suggested that long AU-rich stretches, which are not encountered until after coding regions, are important for restricting polyadenylation to 3′ UTRs (*Lui et al., 2022*). In metazoan mRNAs, an AAUAAA sequence is specifically recognized by the CpA complex (*Chan et al., 2014*; *Schönemann et al., 2014*; *Sun et al., 2018*), and it contributes significantly to determining where polyadenylation occurs. However, given its high frequency in the transcriptome, AAUAAA cannot be the only determinant of poly(A) sites.

The large number of 3′ mRNA isoforms for individual genes indicates that the CpA machinery has relatively low sequence specificity. In addition, as previously noted and shown explicitly here, 3′ isoform endpoints tend to occur in clusters within the 3′ UTR. Such clustering, which is related to microheterogeneity, is usually explained by imprecision of the CpA machinery in the vicinity of a sequence recognition element (e.g. AAUAAA) and/or a preferred cleavage site.

Polyadenylation is intimately connected to the process of transcriptional elongation in vivo (*Nag et al., 2007*; *Pinto et al., 2011*; *Liu et al., 2017*; *Cortazar et al., 2019*; *Goering et al., 2021*), and transcriptional pausing increases polyadenylation in vitro (*Yonaha and Proudfoot, 1999*). An intact RNA tether between RNA polymerase II (Pol II) and the poly(A) site is required for efficient 3′ end processing (*Bird et al., 2005*; *Rigo et al., 2005*). Furthermore, cleavage of the nascent mRNA not only leads to polyadenylation but is also the key step that initiates the processes of transcriptional termination and subsequent export of polyadenylated mRNAs from the nucleus (*Connelly and Manley, 1988*; *Kim et al., 2004*; *West et al., 2004*; *Luo et al., 2006*). In general, each nascent mRNA molecule is cleaved and polyadenylated just once, so the poly(A) profile represents an ensemble of independent events. However, at some human genes, it has been suggested that longer isoforms can be retained in the nuclear matrix and be processed into shorter poly(A) isoforms (*Tang et al., 2022*). In considering the link between elongation and polyadenylation, a key issue is the location of elongating Pol II, and hence, the length of accessible RNA at the time of cleavage and subsequent polyadenylation.

The poly(A) profiles of most yeast genes are altered in cells expressing Pol II derivatives with altered elongation rates (*Geisberg et al., 2020*). In all cases, the same poly(A) sites are used but to different extents. Two slow Pol II mutants each cause a 5′ shift in poly(A) site use, with the slower mutant giving rise to a greater upstream shift. In contrast, each of two fast Pol II mutants causes a 3′ shift in preferred poly(A) sites, although the magnitude of this shift is less pronounced, and fewer genes are affected. These altered poly(A) profiles are due to the Pol II elongation rate because strains with reduced Pol II processivity but normal elongation rates have poly(A) profiles indistinguishable from wild-type strains (*Geisberg et al., 2020*; *Yague-Sanz et al., 2020*). Yeast cells undergoing the diauxic response have poly(A) profiles remarkably like those mediated by slow Pol II mutants, indicating the physiological relevance of Pol II elongation rate to poly(A) profiles (*Geisberg et al., 2020*). Transcription slows down in the vicinity of poly(A) sites, suggesting a functional link between 3′ end processing and elongation (*Parua et al., 2018*; *Cortazar et al., 2019*; *Eaton et al., 2020*). Conversely, elongation rate in metazoans affects alternative poly(A) site choice, with slow Pol II mutants favoring the use of more upstream sites, consistent with a 'window of opportunity' model of poly(A) site choice (*Liu et al., 2017*; *Goering et al., 2021*). However, the fine structure of metazoan poly(A) site clustering and how it is affected by elongation speed have not been investigated.

The shifts in poly(A) profiles in strains expressing fast or slow Pol II mutants could occur gradually or in jumps throughout the 3′ UTR. Here, we address these possibilities by measuring the ratio of 3′ mRNA isoform levels in the speed-mutant vs. the wild-type Pol II in yeast and human cells. Unexpectedly, in both organisms, the mutant:wild-type ratio of isoform expression changes steadily on a nucleotide basis within isoform clusters, whereas it is only minimally changed between clusters. In yeast cells, DNA sequence preferences 13–30 nt downstream of isoform clusters suggest that DNA sequence elements can affect Pol II elongation, subsequent polyadenylation, and the formation of 3′ mRNA isoform clusters. In human cells, Pol II occupancy increases just downstream of poly(A) sites, suggesting a linkage between reduced elongation rate and isoform clusters. Taken together, our results suggest a spatial, and perhaps physical, coupling between the CpA and Pol II elongation complexes, such that cleavage and polyadenylation occur almost immediately upon emergence/ accessibility of the RNA from the Pol II elongation complex.

## Results

### 3′ mRNA isoforms frequently occur in clusters of closely-spaced poly(A) sites

The poly(A) profile of an individual gene is defined by the relative steady-state expression levels of all of its 3′ mRNA isoforms. In previous work, we used the 3′ READS technique to map 3′ mRNA isoforms, and hence poly(A) profiles, in yeast cells expressing wild-type, slow, or fast Pol II derivatives on a transcriptome scale (*Geisberg et al., 2020*). In yeast, 3′ mRNA isoform endpoints occur across a ~200 nt window within the 3′ UTR. Within this 'end zone', visual inspection suggests that isoforms are not randomly distributed but rather appear to occur in clusters of closely-spaced poly(A) sites (*Figure 1A*). For reasons to become apparent, we formalize this observation by considering the likelihood of cluster formation in randomly distributed isoforms for each gene.

In previous work, we defined a 3′ mRNA isoform cluster as a collection of isoforms with closely-spaced 3′ ends and similar half-lives (*Geisberg et al., 2014*). Here, we consider only the spacing between isoform endpoints, defining an isoform cluster as a group of isoforms in which each 3′ endpoint is no more than four nucleotides from the next (*Figure 1A* and *Supplementary file 1*). Our analyses are restricted to 'major isoforms' that are expressed at >5% of the level of the gene's most highly expressed isoform. Major 3′ isoforms account for >97% of overall steady-state mRNA expression. The prevalence of clustered isoform endpoints in each 3′ UTR is far higher than that obtained by randomly distributing the same number of major isoforms over the same window (*Figure 1B*). The same result is obtained when the definition of a cluster is changed by varying the maximal inter-isoform distance from three to seven nucleotides (*Figure 1—figure supplement 1*). As expected, 3′ UTRs containing larger numbers of isoforms give rise to wider clusters but also to more complex cluster patterns that are exceedingly unlikely to be observed by chance (*Figure 1C*). Thus, poly(A) site isoforms occur disproportionately in clusters.

### Distinct patterns of speed-sensitivity within and between clusters in yeast cells

The poly(A) profiles of most yeast genes are altered in yeast strains expressing Pol II derivatives with slow or fast elongation rates (*Geisberg et al., 2020*). Compared to the poly(A) profile in wild-type cells, poly(A) profiles in slow Pol II strains ('slow': F1086S; 'slower': H1085Q) are shifted in an upstream (ORF-proximal) direction, whereas poly(A) profiles in fast Pol II strains ('fast': L1101S; 'faster': E1103G) exhibit subtle downstream shifts. Some genes shift poly(A) profiles in both fast and slow Pol II strains. The Pol II elongation rate has no effect on isoform clustering (*Figure 1—figure supplement 2*).

To address the mechanistic relationship between Pol II speed and poly(A) profiles, we asked whether the shifts in isoform distributions are continuous or occur in jumps throughout the 3′ UTR. For every isoform, we determined its sensitivity to Pol II speed by calculating the ratio of its expression in a Pol II elongation rate mutant (slow or fast) vs. that in a wild-type Pol II strain.

Strikingly, the pattern of isoform ratios in slow vs. wild-type Pol II strains is very different for isoforms within clusters as opposed to isoforms between clusters (two specific examples shown in *Figure 2A*, and transcriptome-scale results shown in *Figure 2B*). Within isoform clusters, both the 'slower':wild-type and the 'slow':wild-type ratios continuously decrease from the most ORF-proximal

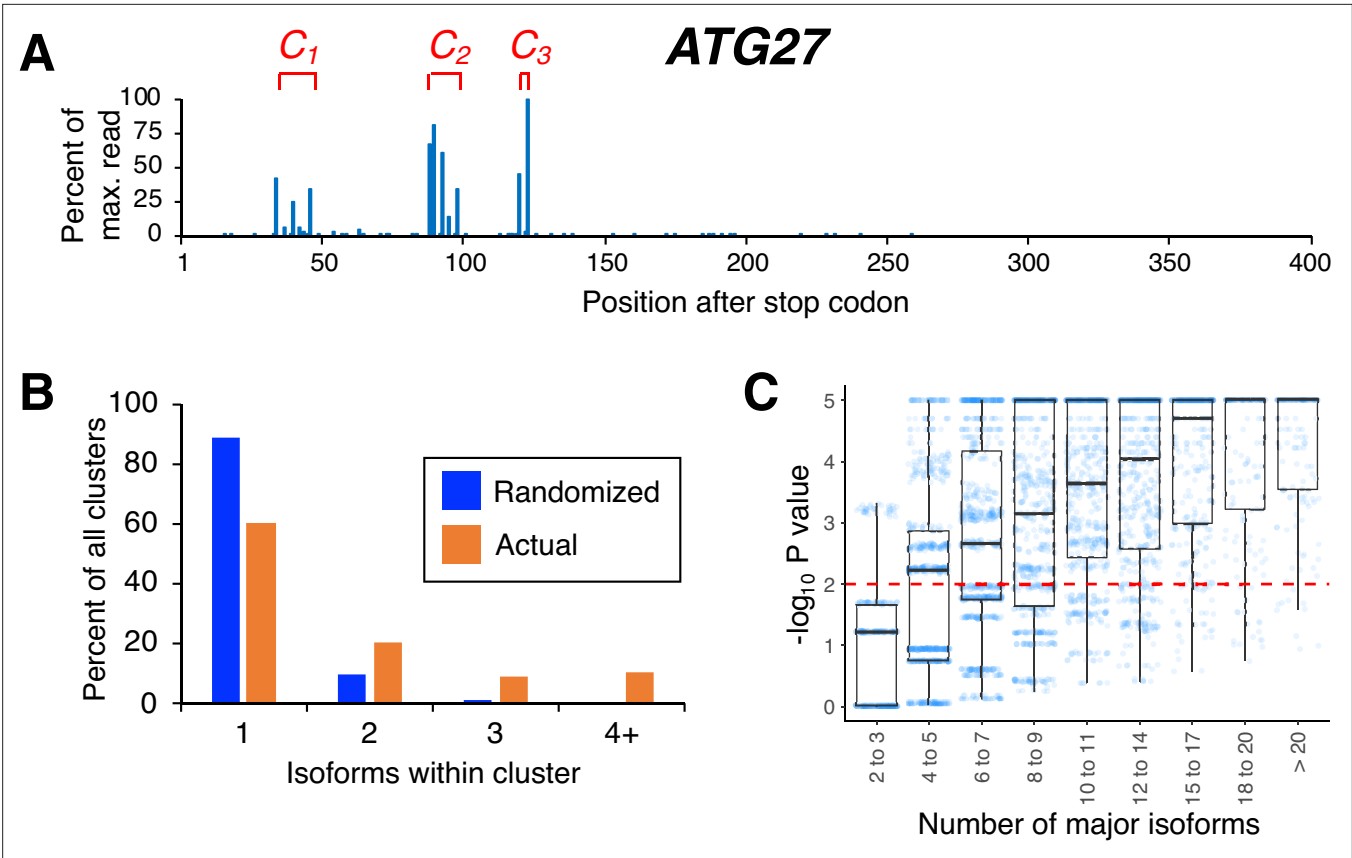

**Figure 1.** Isoforms in yeast 3' untranslated regions (UTRs) are clustered. (**A**) Polyadenylation profile of *ATG27*, a typical yeast gene, illustrating that major isoforms appear in clusters (represented as $C_1$, $C_2$, and $C_3$ in red lettering). (**B**) Frequency distribution of clusters (all isoforms in cluster ≤4 nt apart) containing the indicated number of isoforms in either the randomized or genomic population. The number and frequency of all clusters were tabulated for 3774 genes (orange bars). Potential isoform positions were then shuffled 100,000 times within each gene's 3'UTR, and the frequency and number of isoforms for each cluster were tabulated for every shuffled instance. Cluster frequencies were then combined across all 3774 genes and 100,000 shuffled instances/gene (blue bars). (**C**) Median likelihood ($-\log_{10}$ P value) that the experimentally observed cluster pattern for genes with the indicated number of major isoforms occurs by chance. Each point represents the probability that a given gene's experimentally observed cluster frequency pattern is random. Horizontal bars inside boxplots represent the median values, while the top and bottom of each box represent the 25th and 75th percentiles. Values above dashed red line at $-\log_{10}(P)=2$ are considered statistically significant.

The online version of this article includes the following figure supplement(s) for figure 1:

**Figure supplement 1.** Frequency distributions of naturally occurring and randomized clusters as a function of different cluster definitions.

**Figure supplement 2.** Pol II elongation rate does not affect isoform clustering in 3' untranslated regions (UTRs).

to the most ORF-distal isoforms; i.e., the most downstream isoform within a cluster typically has the lowest slow:wild-type ratio. Remarkably, these decreases in the slow:wild-type ratios occur continuously at the nucleotide level (*Figure 2B*). In addition, the intra-cluster slope is steeper (i.e. the ratio decreases more rapidly) in the strain with the 'slower' Pol II derivative as compared to the 'slow' derivative. In contrast, both slow:wild-type ratios decrease only very slightly for isoforms from the end of one cluster to the beginning of the next cluster, even over large distances (*Figure 2A and B*). These observations do not depend on the maximal inter-isoform distance used to define clusters (*Figure 2— figure supplement 1*).

The same dichotomy of isoform ratios within clusters vs. between isoform clusters is observed for fast Pol II strains, except that the slopes of the 'fast':wild-type and 'faster':wild-type ratios across clusters are positive. Within a cluster, ORF-distal isoforms typically have the highest fast:wild-type expression ratios, with the overall ratios increasing continuously at the nucleotide level (specific example shown in *Figure 3A*, and transcriptome-scale results shown in *Figure 3B*). As observed with the 'slow' and 'slower' Pol II derivatives, the 'faster' Pol II derivative shows a steeper median slope than the 'fast' derivative (*Figure 3B and C*). As with both slow Pol II derivatives, the slope of the ratio change

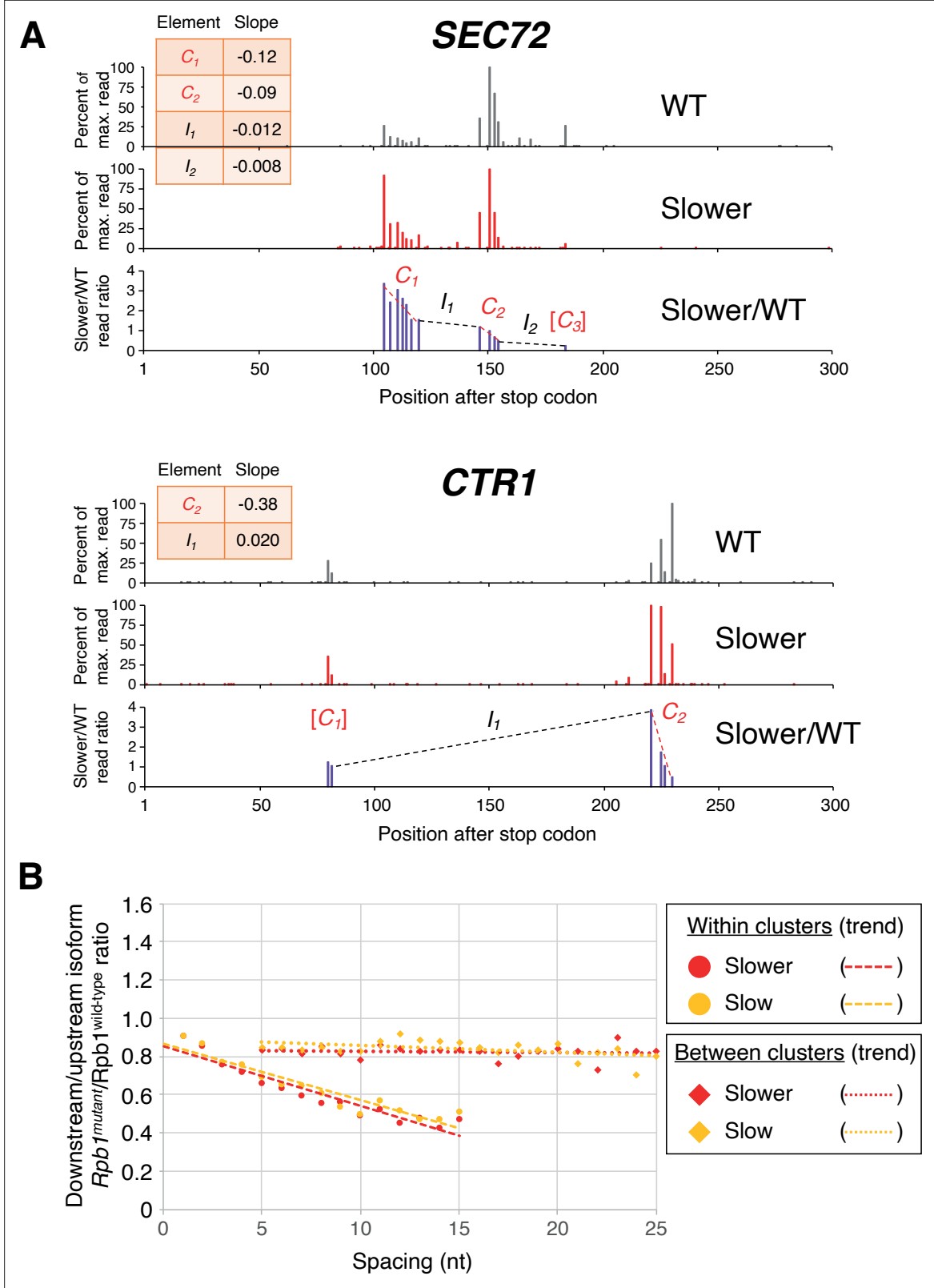

**Figure 2.** Pol II elongation rate drives poly(A) cluster formation. (**A**) Examples of poly(A) profiles in which 'slower'/wild-type (WT) major isoform ratios (purple) decrease more rapidly within clusters than between clusters. Individual isoforms are defined by the number of nt downstream of the stop codon (x-axis). Clusters and inter-cluster regions are depicted as $C_n$ and $I_n$ in red and black lettering, respectively. The subscript $n$ refers to the relative position of either the cluster or the inter-cluster region within the 3' untranslated region, while brackets around clusters indicate that they contain <4

*Figure 2 continued on next page*

*Figure 2 continued*

isoforms and thus were not used in cluster slope analysis. (**B**) Median relative ratios (downstream/upstream isoform) of genome-wide Rpb1(mutant)/Rpb1(WT) utilization at major isoform pairs as a function of nucleotide spacing either within clusters (circles) or in between clusters (diamonds). For each major isoform, Rpb1(mutant)/Rpb1(WT) utilization is computed by dividing the relative expression value of the isoform in the mutant strain by its relative expression in the WT strain. Relative ratios for each isoform pair are calculated by dividing downstream isoform utilization by upstream isoform utilization. Trend lines for 'slower'/WT and 'slow'/WT are depicted via dashes (within clusters) or as dots (between clusters).

The online version of this article includes the following figure supplement(s) for figure 2:

**Figure supplement 1.** The link between Pol II elongation rate and poly(A) cluster formation is independent of exact cluster definition.

is much flatter between clusters. Again, this effect is independent of the precise cluster definition used (*Figure 3* and *Figure 3—figure supplement 1*). Taken together, these results demonstrate a nucleotide-level linkage between Pol II elongation and polyadenylation.

## Mammalian slow Pol II mutant affects poly(A) site micro-heterogeneity within clusters

We compared the polyadenylation profiles in human HEK293 cell lines expressing either an α-amanitin resistant wild-type Pol II or the slow-elongation Pol II derivative with the Rpb1-R749H mutation in the funnel domain (*Fong et al., 2014*). This slow-elongation Pol II derivative often confers an upstream shift in the poly(A) profile resembling that observed in yeast slow Pol II mutants, though occurring at fewer genes (*Goering et al., 2021*). Using 3' READS, we obtained an average of ~30 million reads per biological replicate, with high reproducibility of the data across replicates (*Figure 4—figure supplement 1*).

Analysis of clusters in human cells is more challenging than in yeast due to the greater complexity of the human genome, lower sequencing depth, and the much longer lengths of mammalian 3' UTRs. To work around these limitations, we modified the previous cluster analysis by including all isoforms that contained ≥5 reads in genes with ≤100 reads in the maximally expressed isoform in both wild-type and R749H cell lines, and by defining mammalian 3' UTRs to encompass the region between 1 kb upstream and 5 kb downstream of the consensus stop codon in the Consensus protein coding sequences (CCDS) database. Remarkably, within clusters the median R749H:wild-type ratio exhibits a continuous, nucleotide-level decrease that resembles the decreases observed with both yeast slow Pol II derivatives (an example is shown in *Figure 4A*, and transcriptome-scale results shown in *Figure 4B*; compare *Figure 4B* to *Figure 2B*). As observed in yeast, R749H:wild-type ratios of isoforms from one cluster to the next exhibit much less change (*Figure 4B*). Importantly, the nucleotide-level decrease within clusters observed for R749H:wild-type ratios is independent of both 3' UTR length and the minimal inter-cluster distance used for cluster definition (*Figure 4—figure supplement 2*).

## Cluster-independent analysis of isoform pairs in yeast and human cells

The nucleotide-level link between Pol II elongation rate and polyadenylation is observed only for isoforms within, but not between, clusters. To address the basis of this difference, we performed a cluster-independent measurement of the upstream shift. Specifically, we measured the relative levels of adjacent isoforms in cells expressing slow and wild-type Pol II simply as a function of the distance (in nucleotides) between the isoforms (*Figure 5*). For both yeast slow Pol II mutants and at all distances, the mutant:wild-type expression ratio of the downstream isoform is lower than that of the upstream isoform; the lower the value, the greater the upstream shift. As expected, the 'slower' Pol II mutant confers a greater upstream shift than the 'slow' Pol II derivative. Interestingly, the magnitude of the upstream shift increases slightly with distance at isoform spacings between one and five nucleotides, but it is essentially constant at distances greater than five nucleotides (*Figure 5A*). Similar analysis of human cells expressing the slow R749H vs. the wild-type α-amanitin resistant Rpb1 derivatives yields roughly comparable results, with consistently lower slow:wild-type expression ratios at downstream positions relative to upstream positions within clusters (*Figure 5B*). Thus, in both yeast and human cells, the apparent discordance between slow Pol II effects on isoforms within or between clusters largely reflects the greater distance between consecutive isoforms, not the overall distance traveled by Pol II.

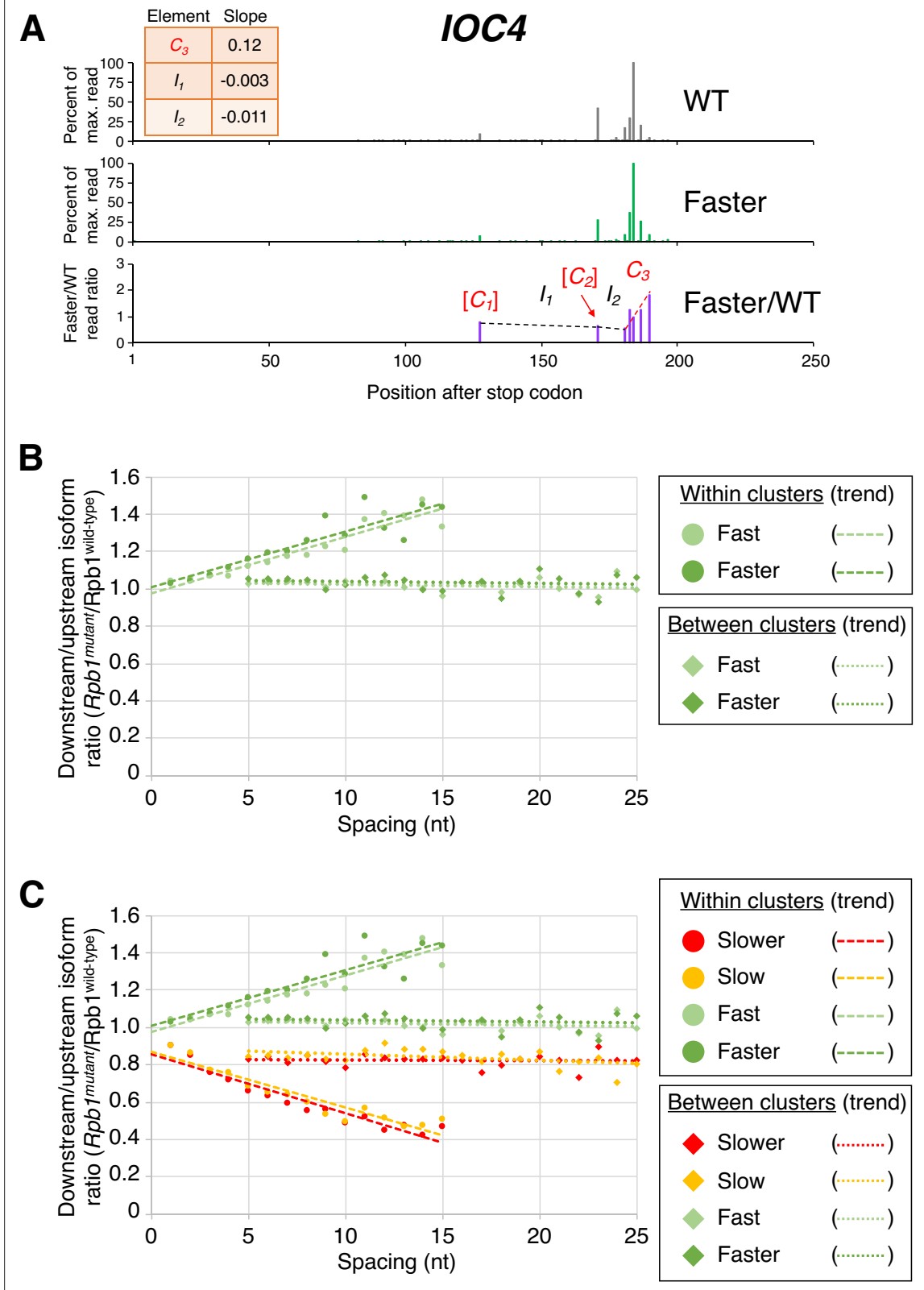

**Figure 3.** Pol II elongation rate drives poly(A) cluster formation. (**A**) Example poly(A) profile in which 'faster'/wild-type (WT) major isoform ratios (purple) increase more rapidly within clusters than between clusters. Clusters and inter-cluster regions are depicted as $C_n$ and $I_n$ in red and black lettering, respectively. The subscript $n$ refers to the relative position of either the cluster or the inter-cluster region within the 3' untranslated region, while brackets around clusters indicate that they contain <4 isoforms and thus were not used in cluster slope analysis. (**B**) Median relative ratios (downstream/upstream

*Figure 3 continued on next page*

*Figure 3 continued*

isoform) of genome-wide Rpb1(mutant)/Rpb1(WT) utilization at major isoform pairs as a function of nucleotide spacing either within clusters (circles) or in between clusters (diamonds). For each major isoform, Rpb1(mutant)/Rpb1(WT) utilization is computed by dividing the relative expression value of the isoform in the mutant strain by its relative expression in the WT strain. Relative ratios for each isoform pair are calculated by dividing downstream isoform utilization by upstream isoform utilization. Trend lines for 'faster'/WT and 'fast'/WT are depicted via dashes (within clusters) or as dots (between clusters). (**C**) Median relative ratios (downstream/upstream isoform) of utilization at major isoform pairs as a function of nucleotide spacing in all four yeast elongation rate mutants ('slower'/WT in red, 'slow'/WT in yellow, 'fast'/WT in light green, and 'faster'/WT in dark green). Relative utilization ratios are depicted as either circles (within clusters) or diamonds (between clusters). Trend lines are dashed for within clusters and dotted between clusters.

The online version of this article includes the following figure supplement(s) for figure 3:

**Figure supplement 1.** The link between Pol II elongation rate and poly(A) cluster formation is independent of exact cluster definition.

## DNA sequence features linked to isoform clusters

Although the nucleotide-level link between Pol II elongation and polyadenylation is based on the behavior of isoform clusters, the results above do not address why 3′ mRNA isoforms occur disproportionately in clusters. Toward this end, we considered the possibility that isoform clusters might form if Pol II encounters a DNA sequence element that slows elongation speed. We were unable to identify any such element spatially linked to yeast cluster formation in general. However, in yeast Pol II mutant strains, increased GC content in the region 13–30 bp downstream of a cluster's most ORF-distal isoform is strongly correlated with more steeply declining (decreasing 'slower':wild-type Pol II and 'slow':wild-type Pol II isoform utilization) and more steeply rising (increasing 'fast':wild-type and 'faster':wild-type isoform ratios) cluster slopes (*Figure 6A*). 'Slower':wild-type Pol II and 'slow':wild-type Pol II clusters display excess GC content at +13 to +30 regions when cluster slopes are highly negative and reduced GC content when cluster slopes are positive (red and orange bars, respectively; *Figure 6B*).

Conversely, 'fast':wild-type Pol II and 'faster':wild-type Pol II clusters exhibit increased GC content at +13 to +30 when their slopes are highly positive and lower relative GC content as the cluster slopes decrease (light and dark green bars, respectively; *Figure 6B*). The contrasting relationships between slow and fast Pol II mutants and GC content downstream of clusters strongly suggest that GC composition at +13 to +30 plays an important role in shaping polyadenylation patterns in clusters by affecting Pol II speed.

Intriguingly, the distance between the +13 and +30 region and the 3′ boundary of an isoform cluster are strikingly similar to the length of the sequence protected by the elongating Pol II machinery (*Bernecky et al., 2016*; *Figure 6C*). This observation suggests the existence of a DNA sequence element that contributes to isoform cluster formation in yeast cells. In human isoform clusters, the lower number of sequence reads did not permit a similar analysis.

## Slower transcription downstream of polyadenylation sites in human genes

Although poly(A) site choice at the nucleotide level is strongly affected by Pol II speed, the relationship between Pol II elongation rate and CpA in the immediate vicinity of poly(A) sites is unknown. We investigated this question by performing eNETseq, a modification of the mNET-seq technique (*Nojima et al., 2015*), that maps the 3′-OH ends of Pol II-associated nascent transcripts, and hence Pol II occupancy, at single base pair resolution in human cells. Reduced Pol II speed at a particular region reflects a longer Pol II dwell time that results in a relative increase in Pol II occupancy within this region.

The composite eNETseq profiles around the region of poly(A) sites show decreasing Pol II occupancy just upstream (region between –40 and –1) of poly(A) sites followed by a biphasic increase in Pol II occupancy after the poly(A) site (*Figure 7A* and *Figure 7—figure supplement 1*). The first increase in Pol II occupancy, which presumably corresponds to a slowdown, occurs 10–25 nt downstream of the poly(A) site. Notably, the poly(A) cleavage site would emerge from the RNA exit channel when Pol II has traveled ~20 nt downstream. This location coincides with the position of the GC-rich region that is linked to isoform clusters in yeast. These observations link CpA with a transcriptional slowdown downstream of the poly(A) site. The second and stronger increase in Pol II occupancy occurs 30–100 nt downstream of poly(A) sites (*Figure 7A*) and appears to be greater in the slow Pol II mutant. It

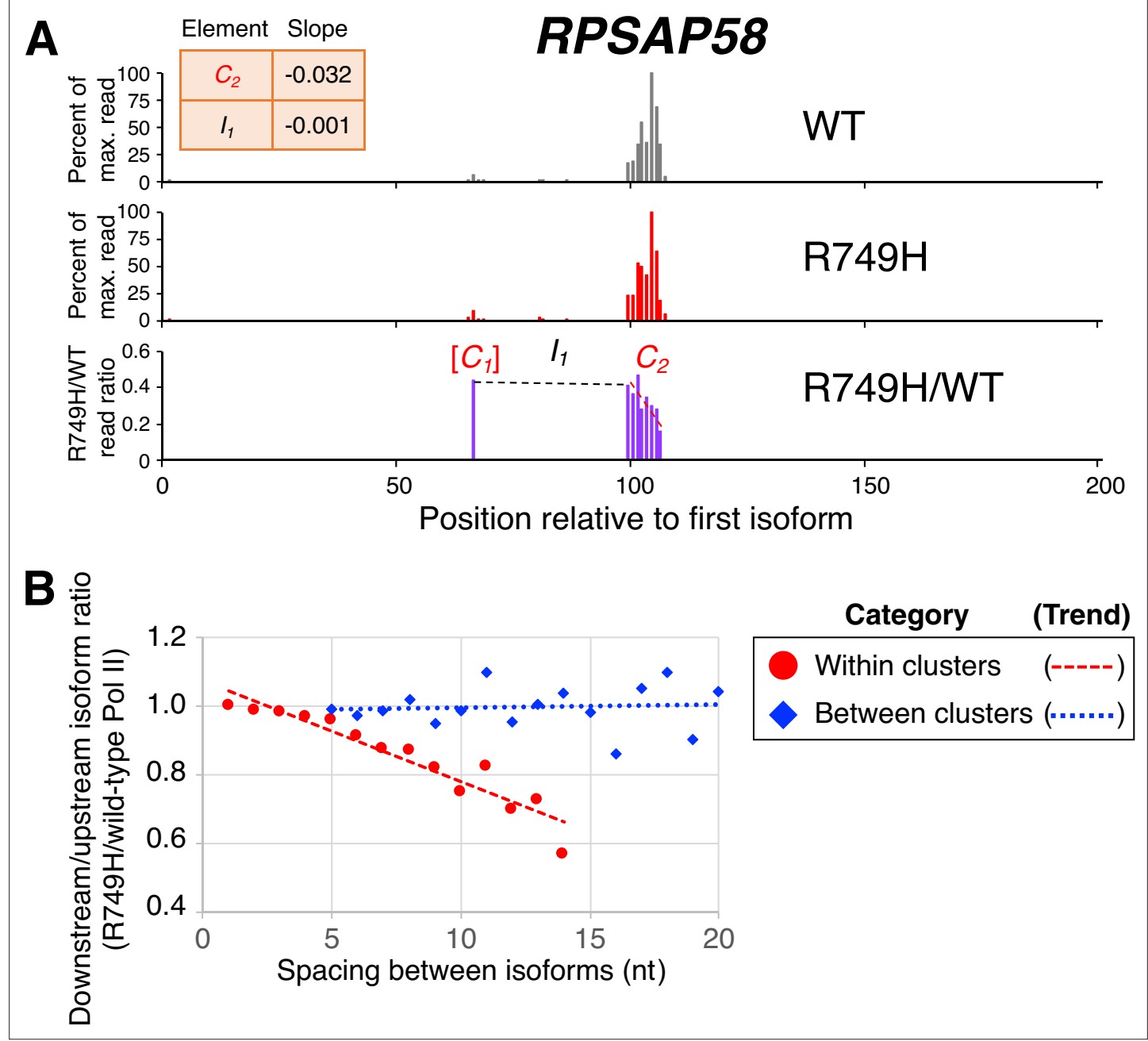

**Figure 4.** Poly(A) cluster formation is also linked to Pol II elongation rate in human cell lines. (**A**) An example of a poly(A) profile in which R749H/wild-type (WT) major isoform ratios (purple) decrease more rapidly within a cluster than between clusters. Clusters and inter-cluster regions are depicted as $C_n$ and $I_n$ in red and black lettering, respectively. The subscript $n$ refers to the relative position of either the cluster or the inter-cluster region within the 3' untranslated region while brackets around clusters indicate that they contain <4 isoforms and thus were not used in cluster slope analysis. (**B**) Median relative ratios (downstream/upstream isoform) of isoform utilization (R749H/WT Pol II) in human 3' isoform pairs as a function of nucleotide spacing. Relative ratios are depicted either as circles (within clusters) or as diamonds (between clusters), while trend lines are either dashed (within clusters) or dotted (between clusters).

The online version of this article includes the following figure supplement(s) for figure 4:

**Figure supplement 1.** Correlation of wild-type (WT) or R749H Pol II biological replicates.

**Figure supplement 2.** The link between Pol II elongation rate and poly(A) cluster formation is independent of the exact cluster definition in mammalian cells.

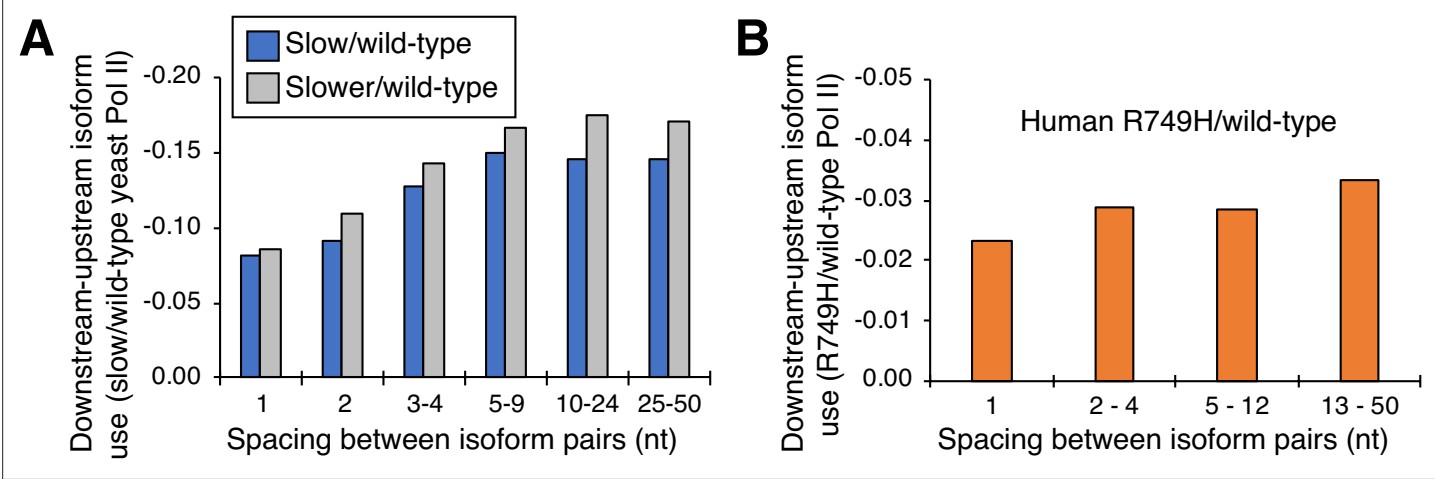

**Figure 5.** Cluster-independent link between Pol II elongation rate and poly(A) formation. (**A**) Median utilization difference (downstream isoform mutant/wild-type expression ratio minus upstream isoform mutant/wild-type expression ratio) is plotted for either 'slower'/wild-type (gray bars) or 'slow'/wild-type (blue bars) as a function of isoform spacing. (**B**) Median utilization difference (downstream isoform R749H/wild-type expression ratio minus upstream isoform R749H/wild-type expression ratio) as a function of isoform spacing.

may reflect even slower Pol II speed during the transcriptional termination process, which involves 5'–3' exonucleolytic degradation of the 3' product that remains associated with elongating Pol II after cleavage (**Nag et al., 2006**; **Cortazar et al., 2019**).

As a control, we examined Pol II localization around intronic AATAAA sequences that are not associated with poly(A) sites (**Figure 7B** and **Figure 7—figure supplement 1**). In contrast to the biphasic increase in Pol II occupancy downstream of poly(A) sites, Pol II occupancy downstream (+30 to +100) of the intronic 'decoy' sites is roughly comparable to occupancy well upstream (−80 to −100). This observation indicates that transcription does not slow down at intronic control sites. For unknown reasons, Pol II occupancy decreases just upstream (positions −40 to −1) of control sites, as it does with poly(A) sites, although we note that there is high AT content in this region around both control and poly(A) sites.

Importantly, Pol II occupancy around poly(A) sites in yeast cells exhibits a remarkably similar pattern (**Harlen et al., 2016**). As in mammalian cells, yeast Pol II occupancy dips just upstream of poly(A) sites and exhibits a marked increase downstream of them (**Harlen et al., 2016**). Thus, the link between recognition and/or processing by the CpA machinery and Pol II slowdown downstream of poly(A) sites appears to be an evolutionarily conserved feature in eukaryotes.

## Discussion

### A nucleotide-level linkage between Pol II elongation and polyadenylation

Both slow and fast Pol II derivatives cause poly(A) shifts within isoform clusters that are continuous at the nucleotide level. This observation strongly suggests a time dependence in which there is a limited opportunity for poly(A) site cleavage to occur at a given site in the RNA before Pol II continues its journey downstream. When the active site of Pol II is at a particular nucleotide location, the longer dwell time of a slow Pol II allows for a longer 'window of opportunity' (**Bentley, 2014**) for cleavage to take place at a potential poly(A) site on the extruded RNA. Faster Pol II, on the other hand, would have a shorter window of opportunity, leading to reduced cleavage at that site.

In general, every mRNA molecule that is cleaved and polyadenylated at a given site can no longer be polyadenylated at any site further downstream. Thus, on a population level, higher polyadenylation frequency at upstream sites means reduced use of downstream poly(A) sites, and lower polyadenylation frequency means increased use of downstream sites. An effect of Pol II elongation speed on cleavage at a given polyadenylation site therefore influences the use of downstream poly(A) sites. The

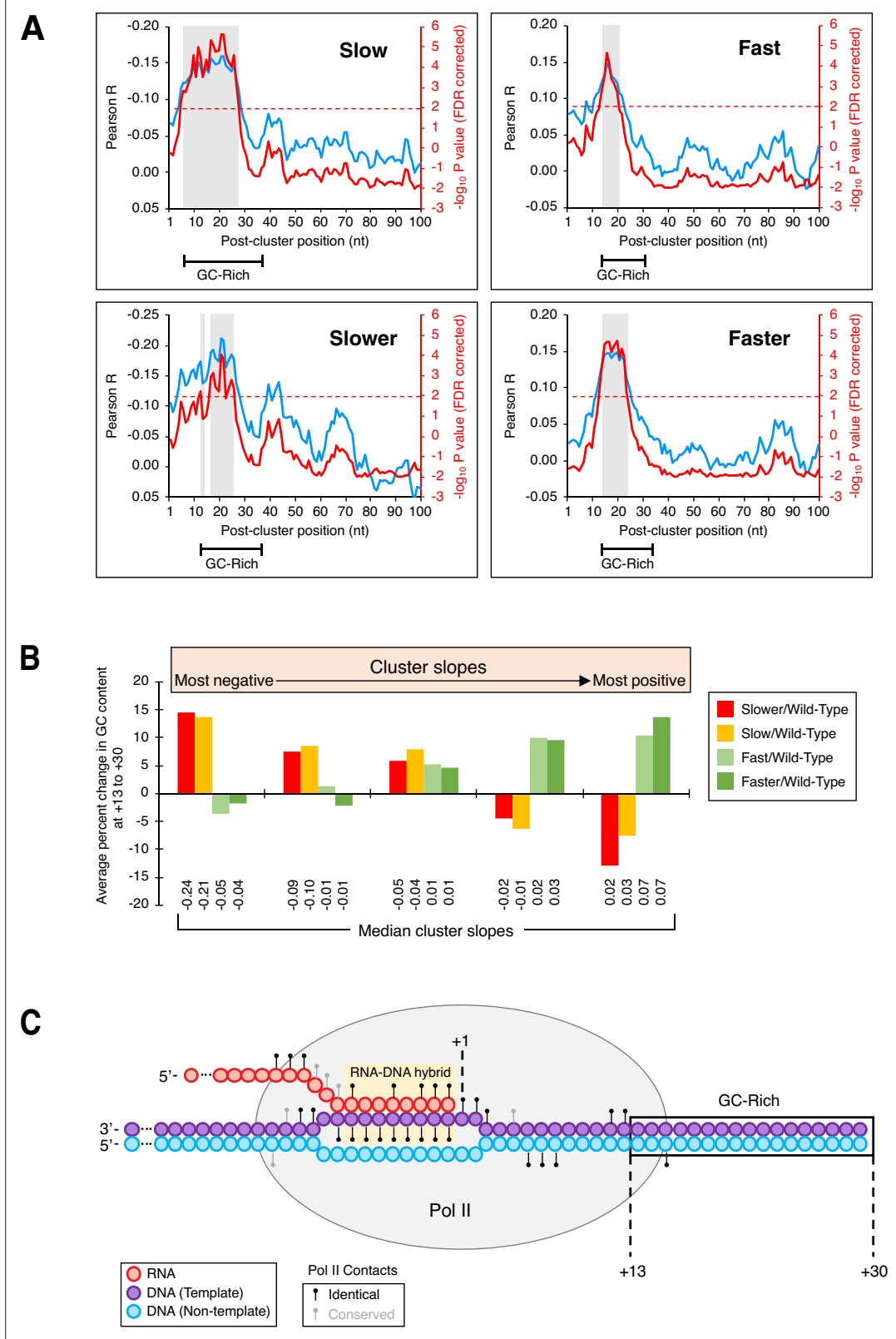

**Figure 6.** GC-rich region just downstream of isoform clusters. (**A**) GC content in a region downstream of clusters is correlated to cluster slopes in 'slow'/wild-type (upper left), 'slower'/wild-type (bottom left), 'fast'/wild-type (top left), and 'faster'/wild-type (bottom left) datasets. Pearson R at each position (blue; left axes) represents the correlation of GC content and cluster slopes in a 10-nt window starting at the position indicated on the x-axis. Red

*Figure 6 continued*

lines (right axes) represent false discovery rate (FDR)-corrected –log₁₀ P values of each correlation, the dashed red line is the significance cutoff (correlations above it are deemed significant), and significant regions are highlighted in gray boxes. Segments at the bottom of each graph indicate the span of the GC-enriched sequence in each mutant/wild-type Pol II dataset. (**B**) GC content at +13 to +30 is linked to cluster slopes. Cluster slopes for 'slower'/wild-type, 'slow'/wild-type, 'fast'/wild-type, and 'faster'/wild-type were individually separated into quintiles, with the most negatively sloped clusters depicted on the left and the most positively sloped clusters depicted on the right. For each cluster, the percent change in GC content at +13 to +30 was computed relative to the median GC content at the equivalent genomic coordinates within 3' untranslated regions. The y-axis depicts the average percent change in GC content for all clusters of a given category within each quintile. Median slopes for cluster categories within each quintile are shown on the bottom. (**C**) GC-rich region immediately downstream of poly(A) clusters in yeast. Elongating Pol II makes numerous contacts (black circles: identical residues in both mammalian and yeast Rpb1; gray circles: conserved residues in both mammalian and yeast Rpb1) with both DNA strands (purple: template strand, blue: non-template strand) and nascent RNA (red). The RNA addition site (+1), Pol II-protected region (gray oval), RNA:DNA hybrid (yellow), and the +13 to +30 region (boxed) are shown. Adapted from *Bernecky et al., 2016*.

conserved relationship between Pol II speed and poly(A) site choice within clusters suggests that Pol II elongation and polyadenylation are mechanistically linked at the nucleotide level.

The Pol II elongation rate has a small, distance-independent effect on the relative levels of isoforms that define the boundaries of adjacent clusters. This indicates that the time-dependent mechanism is linked to 3' end processing and not Pol II elongation per se. Moreover, the upstream shift of isoforms between clusters in a slow Pol II mutant strain is nearly comparable (less than a twofold difference) to that of a single nucleotide within clusters. We do not understand why the magnitude of upstream shift increases slightly, but continuously, with increasing isoform spacings from one to five nucleotides. One possibility is the existence of low-level isoforms arising from inefficient CpA events that occur between major isoforms; these would increase the magnitude of the apparent shift in the same manner as occurs in isoform clusters.

The timing mechanism, although linked to poly(A) site selection, does not address (1) why poly(A) sites occur only at specific positions within the 3' UTR, (2) why levels of 3' isoforms in wild-type cells vary widely and without an obvious pattern (e.g. a simple linear decrease) across a given 3'UTR, and (3) why polyadenylation is very strongly biased to the 3' UTR even though coding regions are typically much longer than 3' UTRs. Thus, the timing mechanism that explains the nucleotide-level link between elongation and CpA operates in concert with specificity elements (e.g. the AAUAAA sequence in mammalian mRNAs) that govern where polyadenylation can occur.

## Evidence that cleavage/polyadenylation occurs soon after the RNA exits the Pol II elongation complex

The nucleotide-level linkage between Pol II elongation rate and polyadenylation suggests that production of a 3' isoform depends on the position of elongating Pol II. Specifically, if Pol II has traveled too far downstream from a polyadenylation site, it is hard to imagine how the Pol II elongation rate would affect activity of the CpA machinery at that site. Once Pol II has traversed the region, the CpA machinery does not go back to the most upstream site but instead tracks with elongating Pol II (*Licatalosi et al., 2002*; *Ahn et al., 2004*; *Glover-Cutter et al., 2008*) and uses sites further downstream within the same cluster. Conversely, if the CpA machinery misses earlier opportunities to act, downstream positions are available to be used as polyadenylation sites. In principle, Pol II speed could affect the poly(A) profile by changing the amount of time for a nascent transcript to adopt a CpA-cleavable pre-mRNA structure before the emergence of more sequence upon Pol II passage makes other structures possible. However, the overall nucleotide-level link strongly suggests a direct connection of poly(A) site choice to Pol II elongation per se.

The nucleotide-level coupling of Pol II elongation to poly(A) site choice strongly suggests that co-transcriptional cleavage of the nascent mRNA can, and often does, occur soon after RNA exits the Pol II elongation complex and becomes accessible to the CpA complex. If, on the other hand, the cleavage reaction was slow relative to Pol II elongation, we would not expect to see an effect of elongation rate on cleavage site selection at single nucleotide resolution when Pol II is located much farther downstream from the poly(A) site. Presumably, the region of newly extruded pre-mRNA is still

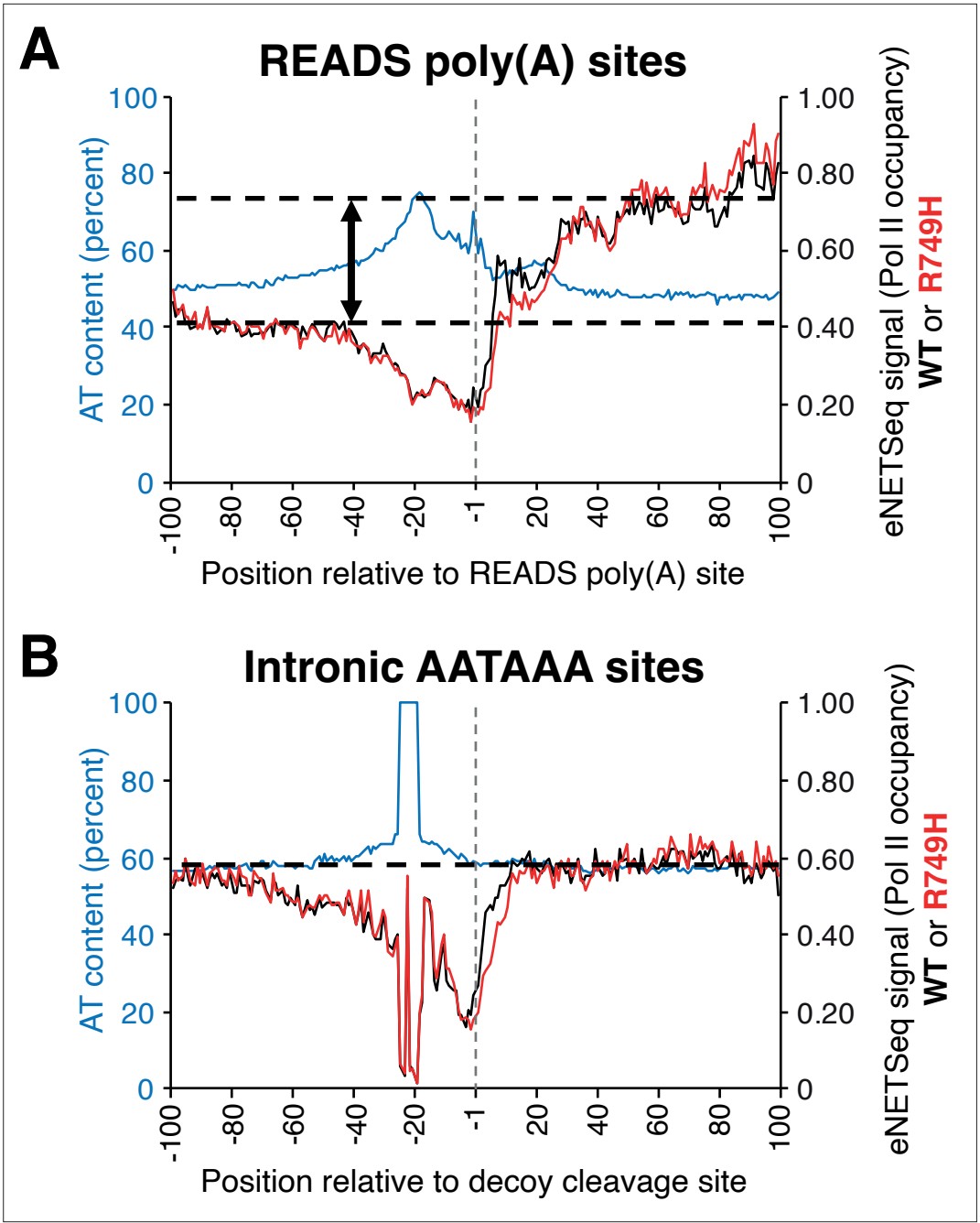

**Figure 7.** Pol II occupancy and AT composition at either READS poly(A) or decoy intronic AATAAA sites in human HEK293 cell lines harboring either wild-type ('WT') or slow ('R749H') Pol II variants. (**A**) eNETseq signal for WT (black; right axis) and R749H (red; right axis) at 2989 READS poly(A) sites (≥20 reads/site in both WT and R749H). Percent AT composition is in blue with the scale on the left axis. The two-sided vertical arrow indicates the difference between the Pol II signal at the region 40-100 nt downstream of the poly(A) site and the average Pol II signal (dashed lines). (**B**) eNETseq signal for WT (black; right axis) and R749H (red; right axis) at 15,865 decoy intronic AATAAA sites (≥10 reads/site in both WT and R749H). Percent AT composition is in blue, with the scale on the left axis. The dashed line indicates the average Pol II signal; note that Pol II occupancy does not increase downstreams of the intronic AATAAA sites.

The online version of this article includes the following figure supplement(s) for figure 7:

**Figure supplement 1.** Correlation of eNETseq biological replicates at poly(A) and decoy intronic AATAAA sites.

in proximity to the elongating Pol II complex, but just far enough away to allow for efficient binding of the CpA machinery and cleavage of the nascent transcript. However, mere RNA accessibility upon exit from the Pol II elongation complex is not sufficient, because local sequences favorable for cleavage by the CpA machinery are also required. These considerations do not preclude cleavage when Pol II has traversed far downstream from the poly(A) site, but such events are very unlikely to be influenced by Pol II speed.

The conclusion that the CpA complex acts soon after RNA exits the elongation complex echoes two other observations. First, co-transcriptional splicing can occur almost as soon as the 3' splice site emerges from the Pol II RNA exit channel (*Oesterreich et al., 2016*). Second, the integrator endo-nuclease complex docks to the RNA exit site and opens to cleave nascent RNA about 20 nucleotides from the Pol II active site (*Fianu et al., 2021*). Notably, several subunits including the catalytic core of the integrator (INTS9/11) and CpA (CPSF100/73) complexes are homologous (*Baillat et al., 2005*; *Elrod et al., 2019*), suggesting that stereo-specific cleavage of the nascent transcript at a preferred position relative to the RNA exit channel might be conserved. In all three co-transcriptional events – splicing, integrator cleavage, and CpA, the processing site may be acted upon almost immediately after the RNA is exposed after Pol II passage.

## Implications for microheterogeneity of poly(A) sites

It has long been observed, and explicitly shown here, that endpoints of 3' mRNA isoforms are clus-tered. The prevailing interpretation of such 'microheterogeneity' of poly(A) sites is that it reflects imprecise cleavage and polyadenylation in the immediate vicinity of a single poly(A) site. Imprecise cleavage occurs in vitro on purified RNA substrates (*Chen et al., 1995*), reflecting the intrinsic prop-erties of the CpA complex. However, the intrinsic activity of the CpA complex on RNA templates, by definition, is independent of Pol II elongation, so imprecise cleavage cannot explain why relative levels of 3' mRNA isoforms within a cluster are strongly affected by the Pol II elongation rate. On the contrary, intrinsic imprecision of the CpA complex should yield the same relative utilization of poly(A) sites in wild-type and mutant Pol II cells. This effect of Pol II elongation rate on neighboring 3' isoform levels also suggests that different poly(A) sites within a cluster largely reflect the stereo-positioning of the CpA complex with respect to the RNA, not intrinsic imprecision of the complex at a single loca-tion. The arguments against imprecise cleavage/polyadenylation dependent on distinct locations of the CpA complex relative to Pol II are particularly strong for isoform clusters that span a large distance.

## DNA sequence elements and Pol II slowdown may contribute to the poly(A) profile via isoform clustering

Although polyadenylation is initiated by RNA cleavage and involves RNA sequence elements in the 3' UTR, the importance of Pol II elongation suggests the possibility that DNA sequence elements might contribute to the polyadenylation profile. For example, a hypothetical DNA sequence element might provide an obstacle to the advancing polymerase, which should favor cleavage and polyadenylation at sites immediately upstream. In yeast, a GC-rich region linked with sensitivity to the elongation rate is located 13–30 bp downstream of isoform clusters and immediately downstream of the DNA bound by the Pol II elongation complex. Thus, we speculate that Pol II has difficulty traversing some GC-rich regions, such that the increased dwell time at these regions leads to increased CpA activity and clus-tering of 3' endpoints in locations where RNA becomes exposed upon Pol II passage.

In this view, nucleotide-level elongation of slow Pol II is more impaired than wild-type Pol II by these GC-rich regions, leading to a greater utilization of the most ORF-proximal isoforms (and a correspondingly faster decline in more distal isoform usage) within a cluster. Conversely, fast Pol II is less impaired by high GC content at +13 to +30, resulting in greater usage of distal isoforms within clusters. Interestingly, both naturally occurring and synthetic GC-rich sequences block elongating Pol II in vitro and in vivo, resulting in increased polyadenylation at upstream sites (*Yonaha and Proudfoot, 1999*; *Yonaha and Proudfoot, 2000*). Whatever the precise mechanism, our results also suggest that, in wild-type cells, DNA sequences that affect Pol II elongation contribute to the formation of 3' isoform clusters.

In human cells, NET-seq analysis reveals increased Pol II occupancy 10–25 nt downstream of poly(A) sites, presumably reflecting decreased Pol II speed in this region. This local decrease in Pol II speed could be due to DNA sequences within this region. Alternatively, it might be caused by recognition

of and/or tighter binding to the AAUAAA (and/or other element) by the CpA machinery, resulting in altered elongation properties of the Pol II machinery. By either mechanism, the reduction in Pol II elongation rate in this region would increase the Pol II dwell time, and hence, contribute to isoform clustering. Whatever the mechanism, increased Pol II occupancy (and hence, Pol II slowdown) is specific to CpA and is not observed at intronic AATAAA sequences.

## A model for the link between Pol II elongation and polyadenylation

As Pol II traverses the gene, there is a continuing decision to elongate further downstream or to cleave and polyadenylate the mRNA, and hence, begin the process of termination. Polyadenylation is largely restricted to the 3′ UTR, yet paradoxically, it occurs to various extents at many sites within the 3′ UTR. The polyadenylation decision depends on both RNA and DNA sequence elements, and it begins when the RNA exits from the Pol II elongation complex and becomes accessible to the CpA machinery.

To explain the nucleotide-level linkage between Pol II elongation and poly(A) site choice, we suggest that the two complexes are spatially, and likely physically, coupled (*Figure 8*). This stereochemical coupling implies that the Pol II elongation and CpA machineries are essentially traveling as a unit, consistent with genome association profiles and Pol II interaction studies of CpA factors (*Licatalosi et al., 2002*; *Ahn et al., 2004*; *Glover-Cutter et al., 2008*; *Carminati et al., 2022*). In this context, cleavage would occur at a constrained (and possibly a fixed) distance from the position of the active site of elongating Pol II.

We imagine the CpA process as occurring in two steps: recognition followed by cleavage of the nascent RNA (*Figure 8*). First, upon transcriptional initiation, Pol II elongation (perhaps together with the CpA machinery) continues unabated until an RNA sequence element(s) in the nascent transcript is exposed and bound by the CpA machinery. RNA sequence elements recognized by the CpA machinery include AAUAAA in many metazoan mRNAs and possibly AU-rich stretches in yeast mRNAs, but other ill-defined sequences are also required for efficient binding such that polyadenylation discriminates efficiently between 3′ UTRs and coding regions. Recognition and hence binding by the CpA machinery to newly exposed sequence elements make the nascent RNA permissive for cleavage and subsequent polyadenylation. More efficient CpA binding functionally couples this complex to the elongating Pol II machinery, possibly slowing Pol II elongation to facilitate the RNA cleavage step.

Second, the coupled CpA and Pol II elongation machineries travel one nucleotide at a time, making a CpA decision at each nucleotide location. In the most extreme version of the model, the nucleotide location of the Pol II active site corresponds to a single nucleotide position where cleavage can occur. In a less stringent model, cleavage occurs within a short window that is determined by the location of the Pol II active site. The level of cleavage and polyadenylation at each position depends on the sequence in the vicinity of that position and the time that Pol II spends at the constrained downstream location. On a population basis, the level of cleavage and polyadenylation at a given position causes a corresponding reduction in the amount of elongating Pol II capable of polyadenylation at more downstream positions. If polyadenylation does not occur at a particular position, Pol II travels to the next position(s), having missed the opportunity for polyadenylation at positions farther upstream. Pol II speed mutants cause changes in the amount of time Pol II spends at each position, thus resulting in nucleotide-level shifts in polyadenylation patterns.

An interesting feature of this model is that CpA at a given site is inefficient, likely due to the short dwell time during which the Pol II elongation machinery is at the constrained location. As a consequence of this constraint, yeast and human genes typically have a large number of 3′ mRNA isoforms. We speculate that the nucleotide-level link between Pol II elongation and polyadenylation evolved to generate multiple 3′ isoforms that have different functional properties.

The above model does not explain the few atypical 3′ UTRs in yeast and human cells in which slow Pol II derivatives lead to a downstream shift, nor the 3′ UTRs and clusters whose poly(A) patterns are unaffected by Pol II elongation rate. In addition, and as discussed above, it does not exclude other mechanisms that could contribute to the poly(A) pattern. However, these alternative mechanisms do not explain the striking nucleotide-level relationship between Pol II elongation and polyadenylation. Thus, our results strongly suggest that the process modeled in *Figure 8* makes a major contribution to the polyadenylation pattern in yeast and human cells.

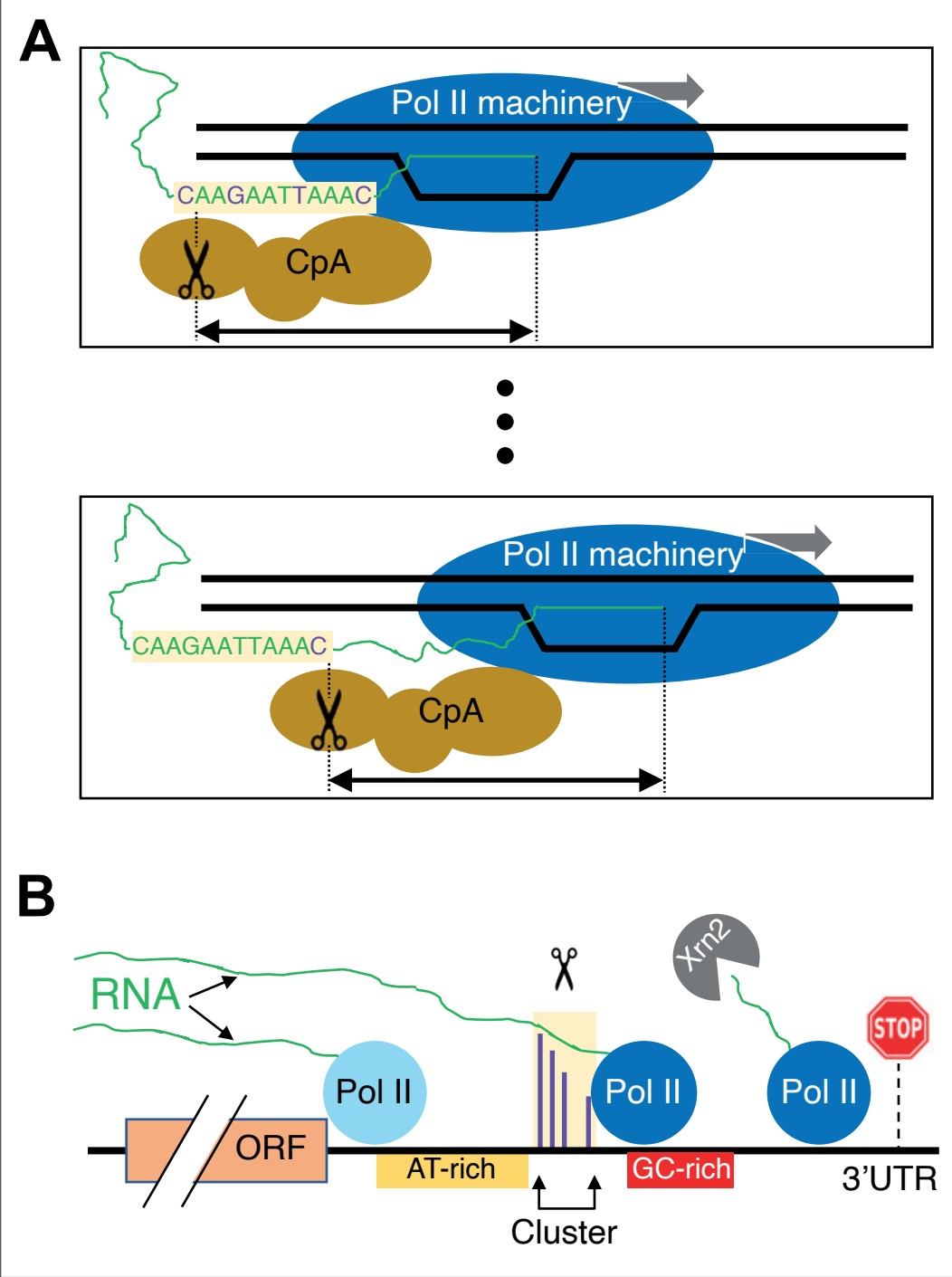

**Figure 8.** Schematic of the link between Pol II elongation rate and poly(A) formation. (**A**) Nucleotide-level link between Pol II elongation and cleavage/polyadenylation (CpA). As the Pol II machinery (dark blue) elongates the nascent RNA one nucleotide at a time, upstream sequences of the newly synthesized RNA strand become exposed, leading the CpA complex (gold) to cleave and polyadenylate the nascent RNA at preferred residues (purple). The Pol II and CpA machineries are spatially, and likely physically, coupled so that cleavage occurs at a constrained distance from the position of the active site of elongating Pol II (black arrow). (**B**) Gene-level view of CpA. Rapidly elongating Pol II (light blue) traverses the gene body and 3' untranslated region (UTR) until it encounters an AT-rich region (gold) and/or a GC-rich stretch downstream of clusters (red), which cause it to slow down (dark blue). The nascent RNA (green) extruded out of the slowing elongating Pol II complex gets cleaved and polyadenylated (scissors) at one of the preferred positions within the cluster (purple). The Pol II complex then

*Figure 8 continued on next page*

*Figure 8 continued*

continues to elongate a short and unstable non-coding RNA that is endonucleolytically degraded by Xrn2 (gray) and eventually terminates at a downstream position (stop sign).

## Materials and methods

### Yeast strains

Parental strain JGY2000 (*MATa his3-Δ0 leu2-Δ0 met15-Δ0 rpb1::RPB1-FRB rpl13::RPL13-FK512*) (*Geisberg et al., 2014*) and Pol II mutant strains JZY5 ('slower'), JZY6 ('slow'), JZY14 ('fast'), and JZY15 ('faster') have been described (*Geisberg et al., 2020*). Sequencing data for all yeast strains were previously published (*Geisberg et al., 2020*) and deposited in GEO (accession #GSE151196).

### Mammalian cell lines

HEK293 Flp-in TREX cells expressing integrated pcDNA5 expression plasmids for WT and slow (R749H) α-amanitin resistant mutants of Rpb1 have been described (*Fong et al., 2014*). 3′ READS libraries were prepared according to *Jin et al., 2015* using RNA extracted after induction with doxycycline (2.0 µg/ml) for 12–16 hr, and treatment with α-amanitin (2.5 µg/ml) plus doxycycline for a further 42 hr.

### Computational processing of datasets

3′ READS data sets were analyzed as previously described (*Geisberg et al., 2020*). In summary, sequence reads were stripped of their initial 4 nt, which consisted of random sequence added during library construction. Any read for which the first non-random nt was not a T, corresponding to an A at the 3′ end of the sequenced mRNA fragment, was discarded. For all other reads, the number of initial Ts (representing the number of terminal As in the sequenced fragment) was appended to the read ID for future reference. These initial Ts were removed, and the read truncated to 17 nt before mapping with bowtie (*Langmead et al., 2009*) to the Sac cer 3 yeast genome build. Next, the sequence adjacent to each mapped read was examined to make sure that the sequenced mRNA fragment contained non-templated As. This was accomplished by comparing each read's previously noted initial T count to the number of As immediately 3′ of its mapped read position in the genome. If the number of neighboring As in the genome was sufficient to account for the initial Ts, the read was discarded as possibly not representing a bona fide polyadenylated product. For each replicate, the remaining read counts were scaled to a total of 25 million.

For sequences derived from human cells, essentially the same procedure was followed, with the following minor differences: after trimming of initial random nt, sequences were stripped of initial Ts and truncated to a maximum of 77 nt. These sequences were mapped using bowtie (*Langmead et al., 2009*) to the hg19 build of the human genome, allowing no mismatches within the first 30 nt of the read. After mapping, only those reads arising from sequences with non-templated As were retained (see above). Sequences mapping to the mitochondrial genome were excluded. For initial comparison of biological replicates, individual replicates (five for wild-type and four for R749H) were each scaled to a total poly(A) read count of 7 million. For other analyses, biological replicates were consolidated, and the combined total scaled to a total of 25 million poly(A) reads. Reads were assigned to a gene if they occurred within 5 kb downstream of the stop codon, as indicated by the CCDS database. Sequencing data for the human cell lines are deposited in GEO (accession #GSE214095).

### Clustering of isoforms and tabulation of clusters in experimental datasets

For each dataset, all genes with ≥1000 reads in 3′ UTRs were analyzed as follows. First, for each gene, all major isoforms (isoforms containing ≥5% of the reads of the maximally expressed isoform) were tabulated, and minor isoforms excluded from further analysis. All major isoforms which were within 4 nt of any other major isoform were then grouped together into a single cluster. Any major isoforms that were located >4 nt away from any other major isoform were classified as single-isoform clusters. Multi-isoform clusters were grouped and plotted according to the number of isoforms per cluster (2 to >20; *Figure 1*, *Figure 1—figure supplement 1*). Clustering and tabulation of clusters with different inter-isoform spacing (i.e. either >4 nt or <4 nt; Figure S1) were performed in an analogous fashion.

## Randomization of major isoform positions and cluster tabulation

For each gene with ≥1000 reads and a minimum of 2 isoforms, we first eliminated all 3' UTR positions at which there are A residues, as it is impossible to distinguish between genomically encoded terminal A's and A residues derived from the poly(A) tail. From the remaining (non-A) positions within the 3' UTR, we then randomly selected the same number of positions as there were major isoforms for the gene. In each random selection, we tabulated the frequencies of clusters with different numbers of isoforms. We repeated the random selection 100,000 times for each gene and computed cumulative frequencies of clusters with different numbers of isoforms. To obtain genome-wide shuffled cluster frequencies, we summed up all gene-specific cluster frequencies for clusters with the same number of isoforms. We then converted the cluster frequencies into percentages by dividing each genome-wide cluster frequency by the total number of clusters and multiplied the resulting fraction by 100 (*Figure 1B* and S1).

To estimate the probability of a gene's isoform pattern being random, we divided the number of times that the cluster frequency pattern (i.e. the frequencies of clusters with specified numbers of isoforms) in each random selection was identical to the experimentally observed cluster frequency pattern. In the event that a gene's experimentally observed cluster pattern didn't appear in any of the 100,000 random selections, the probability was set at p=1/100,000=0.00001. All probabilities were plotted as a function of the number of isoforms per gene for JGY2000 (box plot in *Figure 1C*).

## Comparison of mutant/WT Pol II isoform utilization within and between clusters

Each Pol II mutant dataset was individually analyzed alongside the WT Pol II dataset in the following manner. We limited our analysis to genes with ≥1000 reads in each dataset and then combined the two datasets by including only major isoforms (i.e. isoforms with ≥5% of the reads of the maximally expressed isoform of the same gene in each dataset) that were common to both datasets. We then calculated isoform expression ratios (mutant/WT Pol II) for all the common major isoforms. We compiled a list of clusters in the combined dataset, using 4 nt as the maximum inter-isoform spacing within clusters. In clusters containing four or more isoforms, we used least squares to calculate the slope and Pearson R of the expression ratios vs isoform positions within a cluster. In clusters with high data quality (|R|>0.7), we subdivided all isoform pairs (including isoform pairs that have one or more isoforms between them) into groups based on their spacing (1 nt to 15 nt apart from the first isoform). For each group, we calculated the percent difference between expression ratios (mutant/WT) for each isoform pair (downstream - upstream) and plotted the medians as a function of inter-isoform distance (filled-in circle data points in *Figures 2B, 3B and C*).

Similarly, we sorted inter-cluster regions into different categories based on the distance between two clusters, which we defined as the position of the 5'-most isoform of downstream cluster minus the 3'-most isoform of upstream cluster. For each inter-cluster region, we calculated the percent difference between the expression ratios (mutant/wild-type) for each isoform pair that defines it and plotted the median percent difference in utilization as a function of inter-isoform distance (filled-in diamond data points in *Figures 2B, 3B and C*).

## Mutant/WT Pol II isoform utilization within and between clusters in mammalian cells

The mammalian cell line data (wild-type and R749H) was analyzed in a similar fashion to the yeast datasets, with a few modifications. First, due to the lower sequencing depth, we reduced the minimum threshold requirement of major isoforms to ≥5 reads for genes whose maximum isoform possessed <100 reads in either the wild-type or R749H dataset. Major isoforms were required to possess ≥5% of the reads of the maximally expressed isoform in instances where the latter contained ≥100 reads. Second, we mapped all reads to regions between 1 kb upstream and 5 kb downstream of annotated CCDS termination codons. Clusters were identified using the same definition as above (maximal inter-cluster distance of isoforms = 4 nt), and cluster slopes and associated Pearson R values were computed for all clusters with ≥4 isoforms. In clusters with |R|>0.6, we assigned all possible isoform pairs within the clusters (including isoform pairs that have one or more isoforms between them) into groups based on their spacing (1 nt to 14 nt apart). As in *Saccharomyces cerevisiae*, we computed the percent difference between the R749H/wild-type expression ratios for each isoform

pair (downstream - upstream) and plotted the medians as a function of inter-isoform distance (red circles in *Figure 4B*).

Inter-cluster regions were grouped according to the distance between cluster-bounding isoforms (position of the 5'-most isoform of downstream cluster – position of the 3'-most isoform of upstream cluster). For each inter-cluster region, we calculated the percent difference in the R749H/wild-type expression ratios of the cluster-bounding isoforms that define it and plotted the median percent difference as a function of inter-isoform distance (blue diamonds in *Figure 4B*).

## Cluster-independent analysis of mutant/wild-type isoform utilization as a function of distance

Using the combined yeast ('slower'/wild-type and 'slow'/wild-type) and mammalian (R749H/wild-type) datasets from above, we assigned all neighboring isoform pairs (i.e. any two same-gene isoform pairs that don't have any other major isoforms between them) into groups based on the distance (in nt) between them. For each isoform pair, we computed the difference in downstream - upstream isoform utilization ratios (downstream mutant/wild-type expression ratio minus upstream mutant/WT expression ratio) and plotted the median utilization difference within each group as a function of spacing between isoform (*Figure 5*).

## DNA sequence composition downstream of clusters linked to cluster formation

In each of the *S. cerevisiae* combined datasets ('slower'/wild-type, 'slow'/wild-type, 'fast'/wild-type, and 'faster'/wild-type), we examined the relationship of sequence composition in the vicinity of clusters where data quality surpassed |R|>0.5 to cluster formation. Specifically, we computed the correlations of GC content to cluster slopes in 10-nt sliding windows across sequences either directly upstream (−100 to −1 relative to 5'-most isoform position) or directly downstream (+1 to +100 relative to 3'-most isoform position) of clusters. In each region (−100 to −1 or +1 to +100), we also computed P values for the significance of the correlation at each window position. For the downstream region (+1 to +100), Pearson R (blue; left axis) and multiple hypotheses-corrected p values (red; right axis) were plotted as a function of window position for all four datasets (*Figure 6A*). Analysis of upstream sequences (−100 to −1) yielded R values close to 0, none of which was significant. Likewise, an analysis of GC composition within clusters did not result in any statistically significant R values. Finally, we were unable to find any relationship between cluster slopes and sequence composition in the mammalian data using an identical approach.

To more explicitly show the relationship between GC content at +13 to +30 is and cluster slopes, we first individually separated all cluster slopes for 'slower'/wild-type, 'slow'/wild-type, 'fast'/wild-type, and 'faster'/wild-type into quintiles. Next, we computed the percent change in GC content at +13 to +30 for each cluster by comparing its GC composition at +13 to +30 to the median GC content at the equivalent genomic positions within 3' UTRs. We then plotted the average percent change in GC content for all clusters of a given category within each quintile (*Figure 6B*), with the quintiles ordered from left to right by increasingly positive slope.

## eNETseq Pol II occupancy downstream of mammalian poly(A) sites

eNETseq (*Fong et al., 2022*), a modified version of mNET-seq (*Nojima et al., 2015*) was performed on HEK293 Flp-in TREX cells expressing the wild-type or the slow R749H Rpb1 with N-terminal Avitag after induction with doxycycline (2.0 μg/ml) for 24 hr. The modifications include the following: (1) optimized MNase (NEB) digestion conditions were performed in 50-mM Tris pH 7.9, 5 mM CaCl$_2$, and 250 mM NaCl with 40,000 units/ml for 2 min at 37°C in a thermomixer; (2) after washing the IPs, the beads were treated with a combination of mutant T4 PNK + ATP to phosphorylate 5' OHs and recombinant *Schizosaccharomyces pombe* decapping enzyme Dcp1-Edc1-Dcp2 (75 ng/μl) (*Paquette et al., 2018*) to convert caps to 5' PO$_4$ so that transcript 5' ends can be included in eNETseq libraries; and (3) 12-base unique molecular identifiers were incorporated during library construction to permit unambiguous elimination of duplicates.

eNETseq was performed using ~3–9 × 10$^7$ cells (2–6 15 cm plates) per sample. Nuclei were extracted with 20-mM HEPES pH 7.6, 300-mM NaCl, 0.2-mM EDTA, 7.5-mM MgCl$_2$, 1% NP-40, and 1 M urea prior to solubilization by MNase digestion. Immunoprecipitation was carried out with rabbit

anti-Avitag (Genescript A00674) coupled to protein A Dynabeads. After washing the IPs, on-bead decapping and phosphorylation were performed in a 30 µl reaction with 5 units T4 PNK 3' phosphatase minus (NEB), 2.25-µg GST-Dcp1-Edc1-Dcp2, and 1-µl murine RNase inhibitor in 50-mM Tris HCl pH 7.5, 100-mM NaCl, 5-mM MgCl$_2$, 1-mM DTT, and 0.01% NP-40 at 30°C for 30 min in a Thermomixer. RNA was extracted in Trizol (200 µl), and libraries were generated with QIA miRNA kit seq (Qiagen).

## Pol II occupancy analysis downstream of mammalian poly(A) and intronic decoy sites

eNETseq libraries were sequenced on an Illumina NovaSeq 6000 (2×150). Adapters were trimmed using cutadapt (v2.3), and reads were aligned to the hg38 human genome using Bowtie2 (v2.3.2) (*Langmead and Salzberg, 2012*). PCR duplicates were removed using UMI-tools (v0.5.4) (*Smith et al., 2017*), and read coordinates were collapsed to a single base pair coordinate corresponding to the RNA 3' end. Reads were filtered to only include those with a mapping quality score ≥10, and to remove reads that did not align within 5 kb of a protein coding gene, or that aligned to a snoRNA gene. eNETseq datasets were downsampled so that libraries being compared had the same number of aligned and filtered reads.

To analyze Pol II density in the vicinity of poly(A) sites, we selected all READS poly(A) sites (2989 in total) that contained, in both wild-type and R749H cell lines, ≥20 eNETseq reads within 100 nt of the polyadenylation site and ≥1 read in the first 100 nt downstream of each site. We then computed the percent AT content at each position (–100 to +100) for the 2989 poly(A) sites and plotted the AT composition signal (blue line) alongside the Pol II occupancy in cells harboring wild-type Pol II (black line) or the slow R749H Pol II (red line) (*Figure 7A*).

For the intronic control regions (*Figure 7B*), we first selected all instances of AATAAA in introns >10 kb in length and then filtered out all regions with <10 reads (±100 nt of AATAAA) in both wild-type and R749H Pol II. We then re-centered the AATAAA sequence to –25 to –20 relative to the decoy cleavage site and eliminated any regions that possessed zero reads downstream (+1 to + 100) of the decoy cleavage site. Next, we computed the median AT composition at each position (±100 relative to the decoy cleavage site) in remaining 15,865 decoy intronic AATAAA-containing regions. The median AT composition is depicted in blue alongside the eNETseq Pol II occupancy in cells harboring wild-type (black lines) and R749H Pol II (red lines) (*Figure 7B*).

## Acknowledgements

We thank Catherine Maddox for technical assistance. This work was supported by grants to KS (GM30186 and GM131801) and DB (GM118051) from the National Institutes of Health.

## Additional information

### Funding

| Funder | Grant reference number | Author |
| --- | --- | --- |
| National Institutes of Health | GM30186 | Kevin Struhl |
| National Institutes of Health | GM131801 | Kevin Struhl |
| National Institutes of Health | GM118051 | David L Bentley |

The funders had no role in study design, data collection and interpretation, or the decision to submit the work for publication.

### Author contributions

Joseph V Geisberg, Zarmik Moqtaderi, Conceptualization, Data curation, Software, Formal analysis, Validation, Investigation, Visualization, Methodology, Writing - original draft, Writing – review and

editing; Nova Fong, Formal analysis, Validation, Investigation, Visualization, Methodology; Benjamin Erickson, Data curation, Software, Formal analysis, Validation, Visualization; David L Bentley, Conceptualization, Supervision, Funding acquisition, Project administration, Writing – review and editing; Kevin Struhl, Conceptualization, Supervision, Funding acquisition, Writing - original draft, Project administration, Writing – review and editing

**Author ORCIDs**
Zarmik Moqtaderi (iD) http://orcid.org/0000-0002-2785-7034
Kevin Struhl (iD) http://orcid.org/0000-0002-4181-7856

**Decision letter and Author response**
Decision letter https://doi.org/10.7554/eLife.83153.sa1
Author response https://doi.org/10.7554/eLife.83153.sa2

## Additional files

### Supplementary files
• Supplementary file 1. Supplemental file showing all clusters in YPD.

• MDAR checklist

### Data availability
Sequencing data for the yeast experiment have been previously deposited in GEO (GSE151196). Sequencing for the human experiment has been deposited in GEO (GSE214095).

The following dataset was generated:

| Author(s) | Year | Dataset title | Dataset URL | Database and Identifier |
|---|---|---|---|---|
| Geisberg M, Fong E, Bentley S | 2022 | Nucleotide-level linkage of transcriptional elongation and polyadenylation | http://www.ncbi.nlm.nih.gov/geo/query/acc.cgi?acc=GSE214095 | NCBI Gene Expression Omnibus, GSE214095 |

The following previously published dataset was used:

| Author(s) | Year | Dataset title | Dataset URL | Database and Identifier |
|---|---|---|---|---|
| Geisberg, Moqtaderi, and Struhl | 2020 | The transcriptional elongation rate regulates alternative polyadenylation in yeast | http://www.ncbi.nlm.nih.gov/geo/query/acc.cgi?acc=GSE151196 | NCBI Gene Expression Omnibus, GSE151196 |

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
