## [Editor Report]

Geisberg et al. show, in yeast and human cells, a nucleotide-level relationship between the transcriptional elongation rate and the polyadenylation profile. This suggest that the cleavage/polyadenylation and Pol II elongation complexes are spatially, and perhaps physically coupled so that polyadenylation occurs rapidly upon emergence of nascent RNA from the Pol II elongation complex. Furthermore, the GC-content of sequences downstream of poly(A) clusters influences 3’isoform cluster profiles by slowing down elongation, allowing more time for the 3'-cleavage complex to find the poly(A) site. These findings contribute new information on how the transcription machinery determines which poly(A) site are utilized at the end of genes.

---

## [Decision Letter]

**Decision letter after peer review:**

Thank you for submitting your article "Nucleotide level linkage of transcriptional elongation and polyadenylation" for consideration by *eLife*. Your article has been reviewed by 3 peer reviewers, one of whom is a member of our Board of Reviewing Editors, and the evaluation has been overseen by a Reviewing Editor and James Manley as the Senior Editor. The following individual involved in the review of your submission has agreed to reveal their identity: Bin Tian (Reviewer #1).

Essential revisions:

1) Please address the single major concern noted by Reviewer #1 that relates to calculating cluster difference. This alternative analysis could uncover differences not noted in the initial submission.

2) Using modifications to the text and figures, please address the concerns of clarity raised by Reviewers #2 and #3.

*Reviewer #1 (Recommendations for the authors):*

The authors used the last cleavage site of one cluster and the first cleavage site of the downstream cluster to define usage differences between clusters. Because each cluster can be considered as one cleavage/polyadenylation site, they need to use all sites in each cluster to calculate between cluster differences. In other words, they need to use the sum/median for calculation instead of the last and first site of each cluster. Based on the examples presented in the paper, the sum/median-based calculation should be different than their last-first site calculation, and would probably show some trend for site usage changes between clusters.

*Reviewer #2 (Recommendations for the authors):*

Specific comments

1. Abstract: The meaning of "at the nucleotide level" when referring to the upstream and downstream shifts is not clear to me. Please elaborate.

2. Abstract: The sentence "GC content in a region 13-30 nt downstream from isoform clusters is linked to Pol II elongation rate" must be the other way around.

3. Results: The term "nucleotide-level speed sensitivity" is not clear to me. Explain the meaning of this.

4. Materials and methods: A reference for 3' READS is missing in the section "Computational processing of data sets".

5. Since the speed of elongation of Pol II is central for the data obtained and conclusions drawn, it would be important to actually measure the speed of elongation by the slow, fast, and wt Pol II used in these studies within the genes analyzed. There are techniques estimating elongation rates within all expressed genes using DRB to "synchronize" transcription by loading Pol II onto genes and their pause sites followed by a release and metabolic labeling of nascent RNA. An example of this can be found in Veloso et al. Genome Research 26:896-905, 2014.

6. Figure 1A: It is not clear where the primary data for the frequency distribution of clusters in this figure is located. Please clarify. Are there error bars for the figure? Also, I thought that a "cluster" by definition must contain multiple isoforms so why can one isoform in the figure makes up a cluster? Please clarify.

7. Figure 2A: It would be helpful to have a gene map of the 3' end of gene SEC72 so that we could see where this region of clusters is located. What does the numbering of the X-axis refer to? How is the ratio on the bottom track related to the tracks above? They do not seem to line up for the SEC72 gene but they do for the CTR1 gene. Please explain.

8. Figure 2B: Looks like this data is repeated in Figure 3C where data for both fast and slow Pol II are presented, and it makes more sense there. Thus, Figures 2B and 3B could be deleted.

9. Figure 5: Are there error bars and estimates of significance for these figures?

10. Figure 6A: These data panels are difficult for me to interpret. Are we looking at 3' UTR regions of specific genes? What is the take-home message of these data?

11. Figure 6C: The statement "DNA region linked to cluster formation lies immediately downstream of transcribing Pol II" makes no sense to me since all the sequences within a transcribed gene will at some point be "downstream of transcribing Pol II.

12. Figure 7A&B: The reader is given very little help deciphering the meaning of these figures and what their significance is. Is it that the occupancy of wt or mutant Pol II is similar and that there is less occupancy at AT regions and accumulations of Pol II downstream of poly(A) sites but not in introns following AT-rich regions?

13. Figure 8: These illustrations need to be improved and made more professional looking.

*Reviewer #3 (Recommendations for the authors):*

Overall, I found the study to be an excellent extension of the prior work. Having said this, my main concern is the clarity of the writing and figure presentation. The authors would benefit from clearly describing what is being done (and not relying on the reader to have read their prior paper) and making figures more clearly presented. Below, I provide suggestions.

1. I find figure 1 to be confusing. The text lays out relatively clear definitions of clusters and isoforms and the title of the Figure 1 legend states that yeast 3'UTRs are clustered – yet the data presented in Figure 1 are more focused on frequency and cluster patterns of genes rather than providing the reader with a visual representation of how clusters and isoforms are defined. Since much of the paper relies on histogram measurements of clustering (e.g. Figure 2A), can the authors add a panel in Figure 1 to show representative examples of clustering and isoforms – perhaps disparate representative examples?

2. Along the same lines of clarity, while I appreciate that this is a 'continued findings' manuscript derived from a previous paper, the authors immediately describe isoforms and clusters but fail to actually detail what type of experiment was conducted. The beginning of the results should briefly outline 3READS to make it clear what is being measured (similar to how figure 4 data is initially described but with more detail).

3. The two examples in Figure 2A are confusing to me. The CTR1 gene is crystal clear in my mind and agrees with the wording in the text, however, the SEC72 gene doesn't line up well and it seems that some peaks are ignored in the ratio plot. Also, a minor point, but it might make sense to create a dotted line in the ratio plot from 1.0 to denote unchanged.

4. Equally confusing is why the authors have selected showing two gene examples of the slower mutant compared to wt but are not showing the polyA patterns for the slow mutant for these genes as well. Given that 2b highlights differences at the genome scale, the authors should show gene-specific examples for both strains as well as wt.

5. For figure 2b, it is difficult to assess the statistical differences between slow and slower, which is important because the authors point out differences between the two as meaningful. Can the authors comment on the statistical significance and how that was calculated?

6. Comment #4 is applicable to the fast or faster mutants versus wt shown in Figure 3a. Examples of Fast, faster, and wt should be shown.

7. The authors comment at the end of the figure 4 results that 'importantly, the nucleotide-level decrease within clusters observed for R749…' but no data is shown to support this assertion. Can the authors construct a panel to represent why this claim is being made or simply remove the statement?

8. Figure 6C makes an important point but there is a lost opportunity by the authors to make this point pictorially clearer. The authors could consider having an outline of an RNAPII complex on the portion of the template protected (and increasing the flanking regions of the template in the figure). Also, there is a reference to identical or conserved Pol II contacts in the figure but the Results section makes no reference to this.

9. The authors contend that the occupancy of RNAPII just downstream of the READS PAS (the GC-rich region) of the slow mutant is reduced relative to wt (which I agree with) but the authors state that the 'control' intronic PASs present no such difference – I disagree and observe that the slow mutant has a similarly reduced occupancy in the absence of a GC rich region. Can the authors clarify? Perhaps statistics can be used to support these claims.

---

## [Author Response]

Reviewer #1 (Recommendations for the authors):The authors used the last cleavage site of one cluster and the first cleavage site of the downstream cluster to define usage differences between clusters. Because each cluster can be considered as one cleavage/polyadenylation site, they need to use all sites in each cluster to calculate between cluster differences. In other words, they need to use the sum/median for calculation instead of the last and first site of each cluster. Based on the examples presented in the paper, the sum/median-based calculation should be different than their last-first site calculation, and would probably show some trend for site usage changes between clusters.

The request paper to perform a “sum/median calculation” is based on the erroneous assumption that “each cluster can be considered as one C/P site”. Although previous papers by other groups often condense poly(A) reads within an arbitrarily defined window into a single site, this is *ad hoc* and done for convenience. There is no experimental basis for this assumption.

Furthermore, results in our paper invalidate this assumption. If a single C/P site results in a cluster of poly(A) sites, this would mean the C/P machinery is intrinsically imprecise. In that case, cleavage at that “single site” would be independent of the Pol II elongation rate and hence always give an identical pattern in the “slow”, “fast”, and WT Pol II contexts. Instead, we show that the Pol II elongation rate clearly affects poly(A) site usage at the nucleotide level, which indicates a tight positioning relationship between elongating Pol II and C/P. To put it differently, each poly(A) site is an independent C/P event that is linked to where Pol II is located and its speed

Reviewer #2 (Recommendations for the authors):Specific comments1. Abstract: The meaning of "at the nucleotide level" when referring to the upstream and downstream shifts is not clear to me. Please elaborate.

We clarified that “at the nucleotide level” means from isoform to isoform within clusters.

2. Abstract: The sentence "GC content in a region 13-30 nt downstream from isoform clusters is linked to Pol II elongation rate" must be the other way around.

We specify that GC content…. is correlated to elongation rate without intending to imply any directionality.

3. Results: The term "nucleotide-level speed sensitivity" is not clear to me. Explain the meaning of this.

“Nucleotide-level speed sensitivity” used in the results for the mammalian section, is clearly just a different wording from what was used in the previous section for the yeast results. The results for yeast and human are similar.

4. Materials and methods: A reference for 3' READS is missing in the section "Computational processing of data sets".

We added the missing reference. Thank you for pointing this out.

5. Since the speed of elongation of Pol II is central for the data obtained and conclusions drawn, it would be important to actually measure the speed of elongation by the slow, fast, and wt Pol II used in these studies within the genes analyzed. There are techniques estimating elongation rates within all expressed genes using DRB to "synchronize" transcription by loading Pol II onto genes and their pause sites followed by a release and metabolic labeling of nascent RNA. An example of this can be found in Veloso et al. Genome Research 26:896-905, 2014.

The Pol II mutants used in this paper have been described previously in terms of their elongation rate in vitro and their structures (Kaplan et al., 2012; Braberg et al., 2013). Our previous paper (Geisberg et al., 2020) showed that the 2 “slow” mutants gave remarkably similar phenotypes over the transcriptome, but the effect of the “slower” mutant was slightly more pronounced than the “slow” mutant. Similarly, the 2 “fast” mutants gave remarkably similar phenotypes, with the effect of the “faster” mutant being slightly more pronounced than the “fast” mutant. These published results provide exceptionally strong evidence that the mutants are true Pol II speed mutants, and no further experiments are necessary.

6. Figure 1A: It is not clear where the primary data for the frequency distribution of clusters in this figure is located. Please clarify. Are there error bars for the figure? Also, I thought that a "cluster" by definition must contain multiple isoforms so why can one isoform in the figure makes up a cluster? Please clarify.

We added a new panel (Figure 1A) to clarify the cluster figure and a supplementary file with cluster distributions in individual genes. There are no error bars because this figure just reports aggregated isoform counts. The isoform values represent the consolidation of biological replicates that we previously documented to be extremely highly correlated (R = 0.94; Geisberg et al. 2020). The issue of whether a cluster can contain 1 isoform is semantic. One can still analyze that isoform with respect to other isoforms.

7. Figure 2A: It would be helpful to have a gene map of the 3' end of gene SEC72 so that we could see where this region of clusters is located. What does the numbering of the X-axis refer to? How is the ratio on the bottom track related to the tracks above? They do not seem to line up for the SEC72 gene but they do for the CTR1 gene. Please explain.

We corrected the alignment problem for *SEC72*. The figure does contain a gene map of the 3’UTR but this was unclear because the x-axis label was inadvertently omitted. The figure legend now states that the numbers on the x-axis refer to nt positions downstream of the stop codon.

8. Figure 2B: Looks like this data is repeated in Figure 3C where data for both fast and slow Pol II are presented, and it makes more sense there. Thus, Figures 2B and 3B could be deleted.

Figure 3C is essentially a combination of Figures 2B and 3B, but we thought it useful to have them directly compared. If necessary, we can delete it.

9. Figure 5: Are there error bars and estimates of significance for these figures?

There are no error bars because the values on the y-axis involve combined data from replicates.

10. Figure 6A: These data panels are difficult for me to interpret. Are we looking at 3' UTR regions of specific genes? What is the take-home message of these data?

These plots show that increased GC content downstream of clusters (+13 to +30 relative to the most distal isoform) is linked with the sensitivity of Pol II speed (i.e., extent of upstream or downstream shift) within the cluster. These are not analyses of individual genes, but rather include data from all clusters. The take-home message is that the region between +13 to +30 downstream from the end of a cluster has a GC-rich bias.

11. Figure 6C: The statement "DNA region linked to cluster formation lies immediately downstream of transcribing Pol II" makes no sense to me since all the sequences within a transcribed gene will at some point be "downstream of transcribing Pol II.

In Figure 6C, “transcribing Pol II” refers to the specific location where Pol II is in what is essentially a snapshot. The situation depicted in the figure occurs throughout the transcribed region, but the point of the figure was to show the spatial relationship between Pol II at a specific location relative to the +13 to +30 region.

12. Figure 7A&B: The reader is given very little help deciphering the meaning of these figures and what their significance is. Is it that the occupancy of wt or mutant Pol II is similar and that there is less occupancy at AT regions and accumulations of Pol II downstream of poly(A) sites but not in introns following AT-rich regions?

This is a conventional NET-seq experiment, and the main point of the figure is that there is a big increase in Pol II occupancy (and hence Pol II slowdown) downstream of real poly(A) sites. This increase does not take place at intronic AAUAAA regions that do not support polyadenylation. Elevated Pol II occupancy downstream of poly(A) sites provides strong and independent evidence that polyadenylation and Pol II speed are linked. The WT (black) and mutant (red) Pol II behave similarly to each other, but that isn’t the main point of the figure.

13. Figure 8: These illustrations need to be improved and made more professional looking.

What is meant by “more professional looking?” The figure is like many we have published in the past, and the meaning is clear.

Reviewer #3 (Recommendations for the authors):Overall, I found the study to be an excellent extension of the prior work. Having said this, my main concern is the clarity of the writing and figure presentation. The authors would benefit from clearly describing what is being done (and not relying on the reader to have read their prior paper) and making figures more clearly presented. Below, I provide suggestions.1. I find figure 1 to be confusing. The text lays out relatively clear definitions of clusters and isoforms and the title of the Figure 1 legend states that yeast 3'UTRs are clustered – yet the data presented in Figure 1 are more focused on frequency and cluster patterns of genes rather than providing the reader with a visual representation of how clusters and isoforms are defined. Since much of the paper relies on histogram measurements of clustering (e.g. Figure 2A), can the authors add a panel in Figure 1 to show representative examples of clustering and isoforms – perhaps disparate representative examples?

We added a panel to Figure 1 (see comment 6 of Reviewer 2). The main purpose of Figure 1 is to provide direct computational evidence that clusters exist. Clusters have been previously asserted by inspection, but never formally demonstrated by statistical analysis.

2. Along the same lines of clarity, while I appreciate that this is a 'continued findings' manuscript derived from a previous paper, the authors immediately describe isoforms and clusters but fail to actually detail what type of experiment was conducted. The beginning of the results should briefly outline 3READS to make it clear what is being measured (similar to how figure 4 data is initially described but with more detail).

As suggested, we added 2 sentences at the very beginning of the Results section to describe the previous experiments involving 3’READS.

3. The two examples in Figure 2A are confusing to me. The CTR1 gene is crystal clear in my mind and agrees with the wording in the text, however, the SEC72 gene doesn't line up well and it seems that some peaks are ignored in the ratio plot. Also, a minor point, but it might make sense to create a dotted line in the ratio plot from 1.0 to denote unchanged.

We corrected the alignment problem and x-axis labeling for *SEC72*. Also noticed by Reviewer 2, comment 7.

4. Equally confusing is why the authors have selected showing two gene examples of the slower mutant compared to wt but are not showing the polyA patterns for the slow mutant for these genes as well. Given that 2b highlights differences at the genome scale, the authors should show gene-specific examples for both strains as well as wt.

The difference between the “Slow” and “Slower” mutants was described in detail in our previous paper. This included examples as well as a transcriptome-level analysis. The difference in upstream shifts between the two mutants was obvious upon inspection and was statistically confirmed on a genome-wide level. We don’t think it necessary to show a few examples of both mutants, given that the main point of the figure is the transcriptome-scale analysis

5. For figure 2b, it is difficult to assess the statistical differences between slow and slower, which is important because the authors point out differences between the two as meaningful. Can the authors comment on the statistical significance and how that was calculated?

Within clusters, identical-pair isoform ratios (downstream/upstream read ratios in isoform pairs with sufficient reads in “Slower”, “Slow” and WT Pol II strains) that decrease more rapidly in “Slower”:WT Pol II outnumber those that decrease more rapidly in “Slow”: WT Pol II by > 2.5:1 margin (P = 6.7 x 10^-79^; binomial test). The proportion of “Slower”:WT Pol II ratios that decrease more rapidly is greater than the proportion of “Slow”:WT Pol II that decrease more rapidly for every distance within the cluster.

6. Comment #4 is applicable to the fast or faster mutants versus wt shown in Figure 3a. Examples of Fast, faster, and wt should be shown.

See comment 4.

7. The authors comment at the end of the figure 4 results that 'importantly, the nucleotide-level decrease within clusters observed for R749…' but no data is shown to support this assertion. Can the authors construct a panel to represent why this claim is being made or simply remove the statement?

As requested, we added a new Figure 4—figure supplement 2 to support the statement.

8. Figure 6C makes an important point but there is a lost opportunity by the authors to make this point pictorially clearer. The authors could consider having an outline of an RNAPII complex on the portion of the template protected (and increasing the flanking regions of the template in the figure). Also, there is a reference to identical or conserved Pol II contacts in the figure but the Results section makes no reference to this.

As requested, we modified Figure 6C to have an outline of Pol II covering the protected region; excellent suggestion.

9. The authors contend that the occupancy of RNAPII just downstream of the READS PAS (the GC-rich region) of the slow mutant is reduced relative to wt (which I agree with) but the authors state that the 'control' intronic PASs present no such difference – I disagree and observe that the slow mutant has a similarly reduced occupancy in the absence of a GC rich region. Can the authors clarify? Perhaps statistics can be used to support these claims.

We cannot be certain that the difference between WT and mutant polymerases from +10 to +25 is stronger in real vs decoy poly(A) sites and have edited the text accordingly. Our main point in this figure is to illustrate the dramatic Pol II slowdown farther downstream of real poly(A) sites but not intronic control sites.